# End-to-End AUV Local Motion Planning Method Based on Deep Reinforcement Learning

Xi Lyu [1], Yushan Sun [1,*], Lifeng Wang [2], Jiehui Tan [1] and Liwen Zhang [1]

1   Science and Technology on Underwater Vehicle Laboratory, Harbin Engineering University, Harbin 150001, China; xilyu@hrbeu.edu.cn (X.L.); tanjiehui@hrbeu.edu.cn (J.T.); zhangliwenhrbeu@163.com (L.Z.)
2   Marine Design and Research Institute of China, Shanghai 200011, China; wanglifeng18246@163.com
*   Correspondence: sunyushan@hrbeu.edu.cn

**Abstract:** This study aims to solve the problems of sparse reward, single policy, and poor environmental adaptability in the local motion planning task of autonomous underwater vehicles (AUVs). We propose a two-layer deep deterministic policy gradient algorithm-based end-to-end perception–planning–execution method to overcome the challenges associated with training and learning in end-to-end approaches that directly output control forces. In this approach, the state set is established based on the environment information, the action set is established based on the motion characteristics of the AUV, and the control execution force set is established based on the control constraints. The mapping relations between each set are trained using deep reinforcement learning, enabling the AUV to perform the corresponding action in the current state, thereby accomplishing tasks in an end-to-end manner. Furthermore, we introduce the hindsight experience replay (HER) method in the perception planning mapping process to enhance stability and sample efficiency during training. Finally, we conduct simulation experiments encompassing planning, execution, and end-to-end performance evaluation. Simulation training demonstrates that our proposed method exhibits improved decision-making capabilities and real-time obstacle avoidance during planning. Compared to global planning, the end-to-end algorithm comprehensively considers constraints in the AUV planning process, resulting in more realistic AUV actions that are gentler and more stable, leading to controlled tracking errors.

**Keywords:** autonomous underwater vehicle (AUV); deep deterministic policy gradient (DDPG); deep reinforcement learning (DRL); local motion planning

## 1. Introduction

### 1.1. Background

Motion planning techniques are fundamental to enabling AUVs to navigate autonomously and perform various tasks, attracting significant research interest. Motion planning for AUVs encompasses both global planning and local planning. Global motion planning utilizes known environmental information to devise an optimal route considering distance and safety. On the other hand, local motion planning, guided by the globally planned path, focuses on determining the AUV's position and attitude in real-time based on external environmental information and constraints while avoiding unknown obstacles. In comparison to planning techniques in other domains, local planning for AUVs involves additional constraints, including the complexity of the marine environment and the system dynamics of the AUV itself. Many traditional planning methods fall short in considering AUV constraints and rely solely on environmental characteristics and system dynamics, which hinder adaptability to the environment. It is important to note that AUVs operate not only in open waters but also in narrow spaces such as underwater tunnels, where navigation is severely restricted and multiple wall obstacles impose additional limitations. These factors significantly increase the complexity of local motion planning for AUVs.

Choosing an appropriate planning method lies at the core of AUV motion planning techniques. A well-designed and intelligent planning approach enables AUVs to efficiently and accurately perform planning tasks while avoiding issues like local optima and the curse of dimensionality. Traditional motion planning methods primarily focus on high-level path selection and planning without specifying control commands. However, in recent years, the rapid advancements in reinforcement learning and deep reinforcement learning have led to the emergence of end-to-end motion planning methods. These methods directly map perceptual information to control outputs, integrating the design and implementation of the controller into a single model or algorithm. However, a challenge with end-to-end methods is that mapping control outputs directly, such as thrust, proves to be complex and difficult to learn.

In summary, the selection of appropriate planning methods and their enhancement to suit the AUV's local motion planning system, as well as improving the motion planning capability of AUVs, are crucial prerequisites and research challenges for the intelligent and autonomous development of AUVs. These topics form the primary focus of this study.

### 1.2. Related Work

The safe navigation of robots relies on advancements in the fields of motion planning and control. Extensive research has been conducted in this area, leading to the development of various algorithms for AUV motion planning. Both domestic and international research efforts have produced significant and noteworthy results.

Commonly used motion planning methods include geometric model search, artificial potential field, swarm intelligence, and machine learning. For heuristic search, A* and D* algorithms have been widely used [1,2]. Garau et al. [3] and Jose Isern-Gonzalez et al. [4] both successfully planned complete energy-saving routes by using the A* algorithm, but the heuristic search algorithms are both computationally large and are not applicable to high-dimensional spaces. The artificial potential field proposed by Khatib [5,6] method is a method applied in the field of robot obstacle avoidance where the combined force of gravitational force and repulsive force is calculated as the traction force to make the AUV move. Zhu et al. [7] and Cheng et al. [8] realized path planning under ocean currents and obstacles by combining the artificial potential field with the heuristic method. The artificial potential field algorithm has a well-defined structure and strong obstacle avoidance capabilities, but it also has drawbacks such as local minima. With the rise of artificial intelligence, swarm intelligence bio-algorithms such as the ant colony algorithm and the genetic algorithm have also been applied to the field of motion planning. Bai et al. [9] combined the genetic algorithm with the optimal control theory and solved the time-optimal problem of multivehicle task assignment. Wang et al. [10] applied the improved adaptive ant colony algorithm to the obstacle avoidance problem of AUVs to obtain a shorter collision-free path for AUVs. In addition, swarm intelligence methods also include particle swarm optimization algorithms and firefly algorithms [11–13]. Swarm intelligence algorithms are mainly used for global planning and have the advantages of parallel processing, self-learning, and self-adaptation, but they lack intelligence and real-time capabilities and make it difficult to solve local planning problems.

Machine learning-based mobility planning methods for AUV mainly include neural networks, reinforcement learning (RL), and deep reinforcement learning (DRL) [14]. Xu et al. [15] first proposed a full-coverage neural network algorithm for USV path planning, which can significantly reduce the computation time. Lin et al. [16] proposed a convolutional neural network that generates training and testing datasets in the process of deep learning, which improves the autonomous obstacle avoidance planning ability of the vehicle. The core idea of RL is to use learning as a process of evaluating temptation [17,18] in order to strengthen the autonomy of the agent. Bhopale et al. [19] proposed a Q-learning controller to solve the obstacle avoidance problem of AUVs in unknown environments. El-Fakdi et al. [20] proposed an application scheme for an RL control system for the cable tracking task of an automatic manipulator. Deep reinforcement learning combines the

perception ability of deep learning, which is the current research hotspot of machine learning [21]. Caicedo [22] proposed an underwater robot simulation grid environment based on a kind of DRL end-to-end obstacle avoidance planning algorithm. Cimurs et al. [23], combining DRL and obstacle avoidance navigation systems, obtained the end-to-end training model of the obstacle avoidance system. In 2021, Yu et al. [24] combined imitation learning with the SAC algorithm and proposed an end-to-end motion planning system based on the SAC-GAIL algorithm. The research on AUV motion planning methods based on DRL is still in its infancy, but it is expected to become an effective method for AUV motion planning in the future as the technology evolves. By introducing end-to-end thinking, the mapping problem from action to execution can be solved, and the correct output of the control force can be achieved.

The underwater environment is complex and variable, and the motion of underwater robots can be easily disturbed by currents and other factors, so only focusing on the motion planning technique of AUVs is not enough to achieve safe navigation of AUVs. The merit of an automated control method for AUVs is directly related to whether the AUVs can successfully complete their tasks, which affects the safety performance of AUVs. The development of science and technology has led to more and more control methods being applied to the field of AUVs, such as PID [25], S-plane [26], adaptive [27], sliding mode [28], and other controllers. The design of controllers for marine vessels or robots usually needs to consider their multiple constraint terms [29], suffered from uncertainty disturbances [30,31], as well as control stability, robustness, etc., to design a suitable controller by combining various factors.

Due to its ability to enhance computational efficiency and simplify planning control algorithms, the simplified two-dimensional method has been extensively studied in AUV motion planning. However, notable differences and unresolved challenges persist when comparing it to the planning control of unmanned surface vessels. The underwater environment, in contrast to the surface, presents greater complexity, sensor information delays and absences, and a lack of prominent landmarks for navigation and reward reference, thereby resulting in sparse reward problems when utilizing deep reinforcement learning. Moreover, these factors can introduce interference or noise that compromises the control system, leading to inadequate robot performance. Consequently, underwater robots necessitate higher levels of control stability and robustness. During underwater missions, AUVs must swiftly and effectively make decisions by integrating data to address emergencies, adapt to environmental changes, and fulfill mission requirements. This requirement, in turn, places increased demands on the interconnectedness of sensing, planning, and control.

Upon reviewing the aforementioned research status, it becomes evident that there are existing research gaps in AUV motion planning and control. Traditional planning methods lack autonomy and adaptability, and the problem of sparse rewards in intelligent methods remains unresolved. Additionally, ensuring collision-free paths in cluttered underwater environments necessitates dedicated research to develop effective collision avoidance strategies within confined spaces. Traditional control mechanisms for AUVs suffer from issues such as unstable control, limited robustness, inadequate antijamming capability, and the reliance on precise mathematical models, all of which require improvement through the adoption of intelligent control or hybrid control methods. Most of the current AUV system modules follow a traditional hierarchical pipeline approach, leading to insufficient proximity in information transmission and interaction and the potential for intermodule coupling. In contrast, the end-to-end system, employing deep learning and neural network technology, eliminates the external gap between the perception module and the control module, enabling direct mapping from perception information to thrust output. However, this approach presents challenges, including the complexity of algorithms, high training costs, and difficulties in the learning process.

Therefore, considering the aforementioned research characteristics, we developed an end-to-end local motion planning technique for AUVs based on deep reinforcement learning algorithms in this study. Considering the direct output control method of the end-

to-end training demand is high and learning is difficult, this paper designs a deterministic strategy fetching algorithm based on the double layer depth of the end-to-end perception–planning–control method. The AUV can execute in the end-to-end under the current state of the corresponding action to accomplish tasks. The rest of this paper is organized as follows: Section 2 establishes the end-to-end local motion planning model for AUVs. Section 3 defines the AUV local motion planning method. Section 4 verifies the proposed method using simulation experiments. Finally, we summarize the contributions and limitations of this paper.

## 2. AUV Local Motion Planning Model

Before constructing the local motion planning model, we introduce several assumptions to increase the rigor of the paper. These assumptions are present in theory and simulations, but in practical applications such as field tests, the impact of these assumptions on the real world should be taken into account. The assumptions introduced in this paper and their implications are as follows:

1.  The assumption of accurate sensor data: This study assumes that the sensor data collected by the AUV through sonar is completely accurate and devoid of any noise or uncertainty. However, in reality, various tasks and uncertainties can introduce alterations to the sensor data, which in turn can lead to planning outcomes that do not align with the actual situation. Consequently, the AUV may need to incorporate sensor error correction mechanisms.

2.  The assumption of global information: It is presumed that the AUV has access to comprehensive global environment information, including details such as the mapped environment and the locations and properties of known obstacles. However, in real-world environments, it is more likely that the AUV will have access only to limited local information rather than knowledge of global routes. In such cases, the AUV needs to possess the ability to construct maps synchronously during the search process, which can introduce uncertainty and pose additional challenges.

3.  The assumption of AUV symmetry: This study assumes that the AUV possesses a uniform mass distribution and exhibits symmetry about its X–Z plane. This assumption is made to simplify the dynamics modeling process, reduce the number of parameters required for equations and modeling, and streamline the control algorithm and system analysis. However, it is important to note that this assumption may not hold true when the design of the actual AUV considers specific task and scene requirements. In such cases, a more accurate model that captures the dynamics of the actual AUV is necessary.

4.  The assumption of ideal actuators: It is assumed that the actuators of the AUV can flawlessly execute control commands, regardless of any dynamic response or saturation phenomena. However, in reality, actuators may encounter issues such as delay and saturation, which must be taken into consideration when designing the control system to ensure feasibility and stability.

### 2.1. Kinematic and Dynamic Equation

2.1.1. Kinematic Equation

In this paper, our focus is on studying the underactuated AUV and examining its horizontal motion, encompassing surge, sway, and yaw. To enable a clear description and thorough analysis of the AUV's model, we establish two coordinate systems: the geodetic coordinate system and the AUV coordinate system. These coordinate systems are illustrated in Figure 1.

Vectors $v = [u, v, r]^T$ and $\eta = [x, y, \psi]$, which correspond to the AUV's velocity and position data, can be used to represent the state of the AUV. Therefore, the third degree of freedom kinematic equation is established as follows:

$$\begin{cases} \dot{\eta} = R(\psi)v \\ \dot{\psi} = r \end{cases} \tag{1}$$

$$R(\psi) = \begin{bmatrix} \cos\psi & -\sin\psi & 0 \\ \sin\psi & \cos\psi & 0 \\ 0 & 0 & 1 \end{bmatrix} \tag{2}$$

Here, $\eta$ is the AUV's yaw angle $\psi$ and location $[x, y]$ in the geodesic coordinate system, as well as the AUV's horizontal position vector in the coordinate system; $v$ is the horizontal velocity vector of the AUV in its own coordinate system, including the $X$ axial component $u$, the $Y$ axial component $v$, and the yaw angular velocity $r$. The horizontal motion of the AUV is represented by the three-degree-of-freedom coordinate transformation matrix $R(\psi)$.

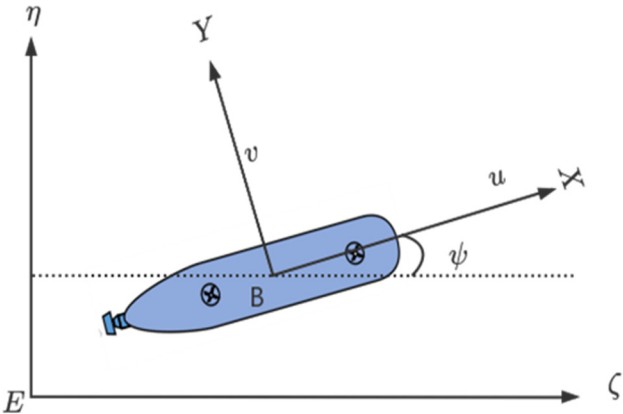

**Figure 1.** AUV coordinate system.

2.1.2. Dynamic Equation

Modeling the dynamics of AUVs is fundamental to the design of control systems and is necessary to study end-to-end motion planning. The control actuators designed in this paper need to control the difference to zero by updating the parameters based on the difference between the target motion parameters output by the dynamic model and the intended target parameters of the planning. Therefore, based on the AUV kinematic equation, we formulate the nonlinear hydrodynamic equation for the AUV as follows [32]:

$$M\dot{v} + C(v)v + D(v)v + g(\eta) = \tau + g_0 \tag{3}$$

In the following, $M$, the system's inertia coefficient matrix, can be divided into two parts: the carriers' inertia matrix $M_{RB}$ and the added mass coefficient matrix $M_A$.

$$M = M_{RB} + M_A \geq 0 \tag{4}$$

The surge, sway, and yaw motions are dissociated as a result of the system's symmetry in its inertia matrix. At the same time, the AUV is assumed to have a uniform mass distribution and to be symmetric with respect to the $x - z$ plane, with moment of inertia $I_{xy} = I_{yz} = 0$ and barycentric coordinate $x_g = 0$, $y_g = 0$. The inertia matrix $M_{RB}$

and the added mass coefficient matrix $M_A$ have the following formulas based on the aforementioned presumptions:

$$M_{RB} = \begin{bmatrix} m & 0 & 0 \\ 0 & m & 0 \\ 0 & 0 & I_z \end{bmatrix} \tag{5}$$

$$M_A = \begin{bmatrix} -X_{\dot{u}} & 0 & 0 \\ 0 & -Y_{\dot{v}} & -Y_{\dot{r}} \\ 0 & -Y_{\dot{r}} & -N_r \end{bmatrix} \tag{6}$$

where the AUV's mass is represented by $m$ and its moment of inertia by $I$, and the rest are the hydrodynamic coefficients. Both $M_{RB}$ and $M_A$ have the following properties:

$$M_{RB} = M_{RB}^T > 0, M_A = M_A^T > 0 \tag{7}$$

$C(v)$ represents the Coriolis centripetal force matrix, which is composed of two parts: Centripetal force coefficient matrix $C_{RB}$ and Coriolis force coefficient matrix $C_A(v)$:

$$C(v) = C_A(v) + C_{RB}(v) \tag{8}$$

where:

$$C_A(v) = \begin{bmatrix} 0 & 0 & Y_{\dot{v}}v + Y_{\dot{r}}r \\ 0 & 0 & -X_{\dot{u}}u \\ -Y_{\dot{v}}v - Y_{\dot{r}}r & X_{\dot{u}}u & 0 \end{bmatrix} \tag{9}$$

$$C_{RB}(v) = \begin{bmatrix} 0 & 0 & -mv \\ 0 & 0 & mu \\ mv & -mu & 0 \end{bmatrix} \tag{10}$$

$C(v)$ is an antisymmetric matrix, $C_{RB}(v) = -C_{RB}(v)^T$, $C_A(v) = -C_A(v)^T$.

The AUV's hydrodynamic damping matrix, denoted by $D(v)$, is defined as follows:

$$D(v) = - \begin{bmatrix} X_{u|u|}|u| & 0 & 0 \\ 0 & Y_{v|v|}|v| + Y_{v|r|}|r| & Y_{r|v|}|v| + Y_{r|r|}|r| \\ 0 & N_{v|v|}|v| + N_{v|r|}|r| & N_{r|v|}|v| + N_{r|r|}|r| \end{bmatrix} \tag{11}$$

Since only the plane motion of the AUV is considered in this study, $g(\eta)$ stands for the restoring force/moment vector, which is ignored.

The control input vector is represented by $\tau$. The AUV used in this study is underactuated, and there are fewer system inputs than degrees of freedom of motion. Specifically,

$$\tau = \begin{bmatrix} \tau_u & 0 & \tau_r \end{bmatrix} \tag{12}$$

where, respectively, $\tau_u$ and $\tau_r$ stand for the surge force and yaw moment.

$g_0$ stands for the disturbance vector, which is assumed in this research to be zero for the sake of the study.

### 2.2. Sensor Model

Figure 2 illustrates the construction of the perception model for the AUV sensor. It is equipped with nine single-beam obstacle avoidance sonars, which are used to sense the position of obstacles and the straight-line distance between the AUV and the obstacles. These sonars are denoted by the symbols S1–S9. The obstacle avoidance sonars have a sampling frequency of 2 Hz and can detect objects up to a distance of 120 m.

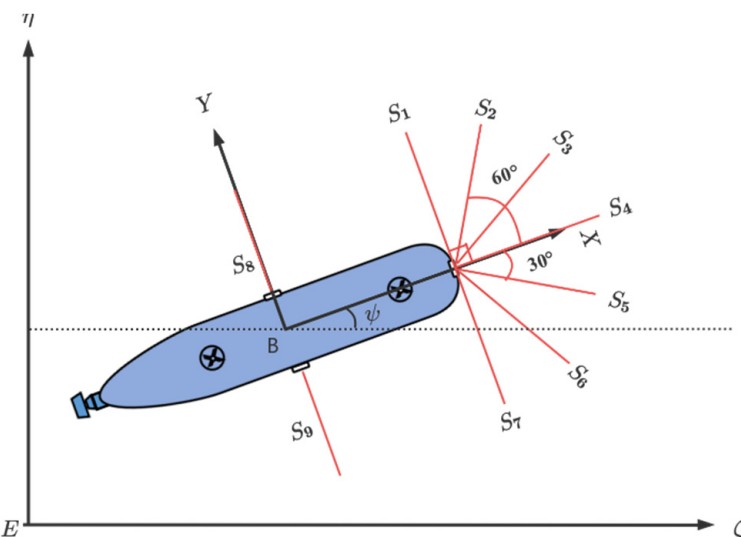

**Figure 2.** AUV sensor model.

*2.3. Local Motion Planning System*

2.3.1. End-to-End Planning System Framework

The underwater operation of an AUV involves a complex process. When faced with multiple constraints in an AUV task, relying solely on global path planning– control mapping is impractical. This approach overlooks many real-world situations and constraints that arise during intermediate stages, leading to inaccurate control force constraints. To enhance the autonomous obstacle avoidance decision-making capability of AUVs in locally observable underwater environments, local planning is introduced. This approach considers the environmental characteristics and task requirements of AUV operations. By combining local planning and control, an end-to-end local motion planning framework is established. This framework encompasses perception, planning, and control. Within this framework, local motion planning is guided by the global path node as the target point. Real-time perception information is obtained through sensors, enabling the AUV to respond to emergencies while tracking the global path.

The planning system presented in this paper combines motion planning with control execution. To address the learning difficulty of the algorithm within the system, an end-to-end approach based on the two-layer deep deterministic policy gradient algorithm is proposed. This approach involves establishing a state set based on environmental information, an action set based on the AUV's motion characteristics, and a control force set based on control constraints. The reward value function is defined using the multi-constraint model of AUV motion, facilitating the construction of a mapping between state–action–control forces. The mapping between state and action represents the planning process, while the mapping between action and control forces represents the execution process, specifically the control process of the AUV. By training the mapping relationship between each set, the end-to-end perception–planning–execution method is realized. The specific framework of the end-to-end planning system is illustrated in Figure 3.

2.3.2. Constraint Model

The motion planning of the AUV is influenced by various dynamic and kinematic constraints inherent to the vehicle. These constraints encompass factors such as the AUV's motion speed, acceleration, rotation angle, force, and other dynamic and kinematic characteristics. When aiming to achieve global path tracking and real-time obstacle avoidance through local motion planning, the following constraints are given primary consideration (Table 1):

**Table 1.** Constraints for AUV motion planning.

| Constrained Type | Constraint Condition |
|---|---|
| Distance between the AUV and obstacles | It is stipulated that the distance between the AUV and the obstacle is not less than 5 m, which means: $d_{1-9} \geq 5$ m. |
| Maneuverability constraint | Considering the mobility constraint, the angle transformation of AUV in each step is limited to be no more than 6°, that is, $\theta\prime \leq \theta + \pi/30$. The maximum difference between the target heading angle and the current heading angle is limited to $\pi/6$. |
| Distance between the AUV and the path target point. | Let $L_{target}$ denote the distance from the AUV to the goal point of the path, and when $L_{target} \leq 2$ m, the AUV is considered to have reached the goal point. |
| Constraints on the length of the navigation path | The path length holds considerable significance in influencing the AUV motion process, and it directly affects the path time as well. Consequently, it is crucial to minimize the path length and reduce the path time during the planning process. To achieve this objective, the introduction of a constraint on AUV roaming becomes necessary. |

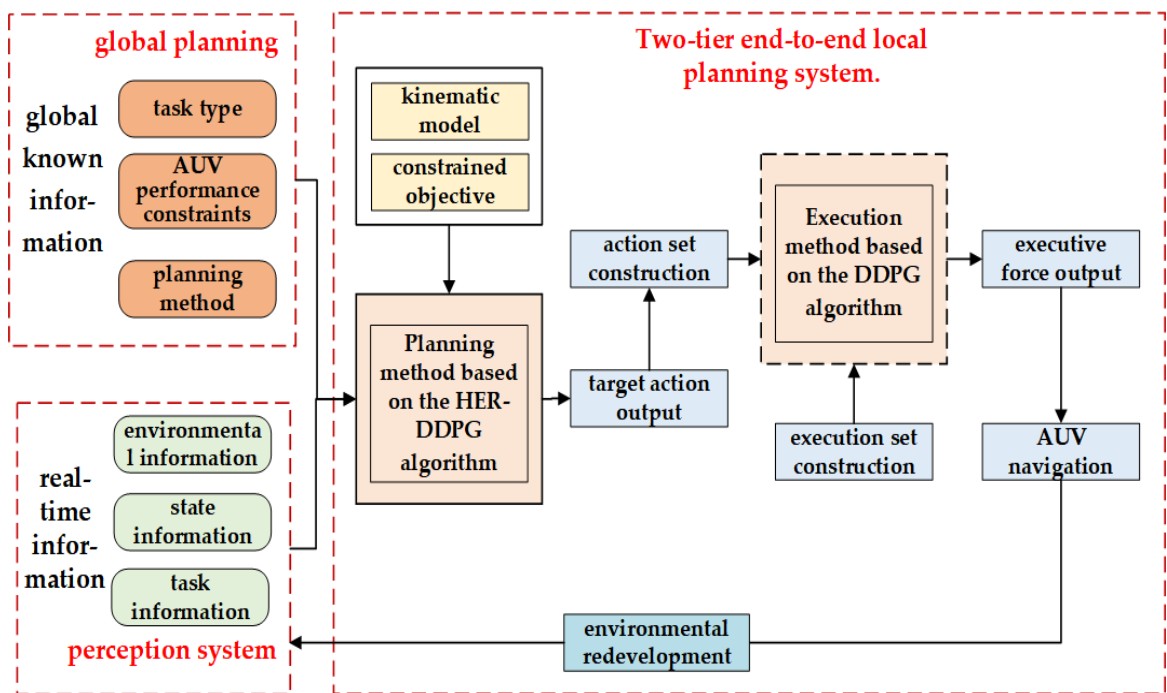

**Figure 3.** The end-to-end planning system framework.

## 3. Method

### 3.1. Deep Reinforcement Learning Algorithm

3.1.1. MDP Model of AUV Path Planning

A Markov decision process (MDP) [33] model needs to be created by fusing the constructed environment with the gathered data in order to undertake deep reinforcement learning training. A quadruple $(P, S, A, R)$ makes up the MDP model, where $A$ stands for the action space, $S$ for the state space, $P$ for the likelihood of a state change, and $R$ for the reward function. The planning system process set using the MDP model can be understood as follows: after receiving the state information $S_t$ at time $t$, the AUV outputs the action

$a \in A$ under the action of policy $\pi$, the state changes to $S_{t+1}$, and the reward value $r_t$ at time $t$ is received at the same time.

### 3.1.2. DDPG

The deep deterministic policy gradient (DDPG) [34] algorithm was created by the Google DeepMind team, employing the actor–critic (AC) algorithm framework, the experience replay mechanism, and a dual network structure construction technique. This algorithm has exhibited exceptional performance in addressing Markov sequence decision problems within a continuous action space.

In the DDPG algorithm, the deterministic policy is the deterministic action value $a_t$ taken in state $s_t$ at time $t$, and the reward value function is the expected reward after taking action $a_t$ in the state $s_t$. The action-value function $Q(s, a|\theta^Q)$ is denoted by policy network with parameter $\theta^\mu$, the deterministic policy $a = \mu(s_t|\theta^\mu)$, and by value network with parameter $\theta^Q$.

Assuming an agent's initial state is $s^1$, and the state distribution follows $\rho^\mu$ under a deterministic policy $\mu$, then the objective function defined in the DDPG algorithm is as follows:

$$J(\theta^\mu) = E_{s\sim\rho, a\sim\mu}[R_1] \tag{13}$$

In the DDPG algorithm, the reinforcement signal is defined as the future discounted reward total return $R_t$, which represents the integrated evaluation of the agent's behavior action decision strategy. The method is as follows:

$$R_t = \sum_{i=t}^{T} \gamma^{(i-t)} r(s_i, a_i) \tag{14}$$

In the above equation, $\gamma$ is the discount factor, indicating that more distant rewards have less impact on the evaluation of the current state, and $r(s_i, a_i)$ represents the reward value obtained by choosing action $a_i$ in state $s_i$.

The policy that seeks to maximize the objective function $J(\theta^\mu)$ is defined as Equation (15):

$$\mu^* = \underset{\mu}{\operatorname{argmax}} J(\theta^\mu) \tag{15}$$

Gradient descent is used to optimize the objective function:

$$\nabla_{\theta^\mu} J = E_{s_t\sim\rho^\beta}\left[\nabla_a Q\left(s, a\middle|\theta^Q\right)\Big|_{s=s_t, a=\mu(s_t)} \nabla_{\theta^\mu} \mu(s|\theta^\mu)_{s=s_t}\right] \tag{16}$$

In Equation (16), $E_{s_t\sim\rho^\beta}$ represents the expected value of the value function $Q$ of state $s_t$ following distribution $\rho^\beta$.

For the critic network, minimize the loss to update the parameters, as shown in Equation (17).

$$L(\theta^Q) = E_{s_t\sim\rho^\beta, a_t\sim\mu, r_t\sim E}\left[(Q_{Target} - Q(s_t, a_t|\theta^Q))^2\right]$$
$$Q_{Target} = r(s_t, a_t) + \gamma Q'\left(s_{t+1}, \mu(s_{t+1}|\theta^\mu)\middle|\theta^{Q'}\right) \tag{17}$$

The DDPG algorithm uses the DQN algorithm to calculate the parameter gradient of the deep convolutional neural network model, and updates the critic network according to Equation (18):

$$\frac{\partial L(\theta^Q)}{\partial\theta^Q} = E_{s_t\sim\rho^\beta, a_t\sim\mu, r_t\sim E}\left[(Q_{Target} - Q(s_t, a_t|\theta^Q))\nabla_{\theta^Q}(s, a|\theta^Q)\right] \tag{18}$$

The target policy network parameters $\mu'$ and target value network parameters $Q'$ are updated by the DDPG algorithm using the experience replay approach. By retrieving data

from the experience replay pool and storing the quadruple $(s_t, a_t, r_t, s_{t+1})$ there, the target network parameters are updated:

$$\begin{cases} \theta^{Q'} \leftarrow \tau\theta^Q + (1-\tau)\theta^{Q'} \\ \theta^{\mu'} \leftarrow \tau\theta^Q + (1-\tau)\theta^{\mu'} \end{cases} \tag{19}$$

where $\tau$ is the update coefficient, which is usually small, such as 0.1 or 0.01.

### 3.2. Planning Method Based on the HER-DDPG Algorithm
### 3.2.1. HER-DDPG

When DDPG algorithms are utilized for goal-oriented tasks such as path planning, the rewards tend to be sparse, leading to slow overall training of the algorithm. To address this issue, the hindsight experience replay (HER) algorithm [35] was introduced at the 2017 Neural Information Processing Systems (NeurIPS) conference and has since gained widespread acceptance as a solution to the sparse reward problem. The HER-DDPG algorithm, obtained by combining the DDPG algorithm with the hindsight experience replay method, offers notable advantages. Unlike the original DDPG algorithm, the HER-DDPG algorithm achieves denser reward signals by resetting goals for historical trajectories. It then leverages the original failed data to extract successful experiences relevant to the "new task," resulting in improved training stability and sample efficiency.

With the hindsight algorithm, the reinforcement learning agent is trained by treating the achieved goal as a virtual goal, recalculating the reward and putting it into the experience pool. Suppose now that a policy is used to explore the environment with goal $g$, resulting in a trajectory such that $s_1, s_2, \cdots, s_T$ and $g \neq s_1, s_2, \cdots, s_T$. This means that we receive a reward of $-1$ for the entire trajectory, which is extremely little for our training. After the experience replay, the entire trajectory is reexamined with a different goal $g\prime$. Although the goal $g$ is not reached, the policy completes the goal corresponding to $s_1, s_2, \cdots, s_T$, that is, completes the goal $\varphi(s_1), \varphi(s_2), \cdots, \varphi(s_T)$ during the exploration process. If we use these goals to replace the original goal $g$ with a new goal $g\prime$, and recalculate the reward values in the trajectories, we can provide the policy with useful training information from the experience of failure. The specific algorithm process of hindsight experience replay is as follows (Algorithm 1):

---

**Algorithm 1** HER algorithm

---

1: Set up experience replay pool $R$ and policy $\pi$'s parameter $\theta$'s initial values.
2: For episode $e = 1 \rightarrow E$ do:
3:    According to the goal $g$ and initial state $s_0$ given by the environment, the trajectory $\{s_0, a_0, r_0, \cdots, s_T, a_T, r_T, s_{T+1}\}$ is obtained by using $\pi$ to sample in the environment, and it is stored in $R$ as $(s, a, r, s', g)$.
4:    Sample N $(s, a, r, s', g)$ tuples from R
5:    Choose a state $s''$ for these tuples, map it to a new objective $g' = \varphi(s'')$, calculate a new reward value $r' = r_{g'}(s, a, s')$, and then replace the old tuples with the new ones using the new data $(s, a, r, s', g')$.
6:    This new set of tuples is used to train policy $\pi$.
7: End for

---

The following Figure 4 illustrates the precise neural network structure of the HER-DDPG algorithm:

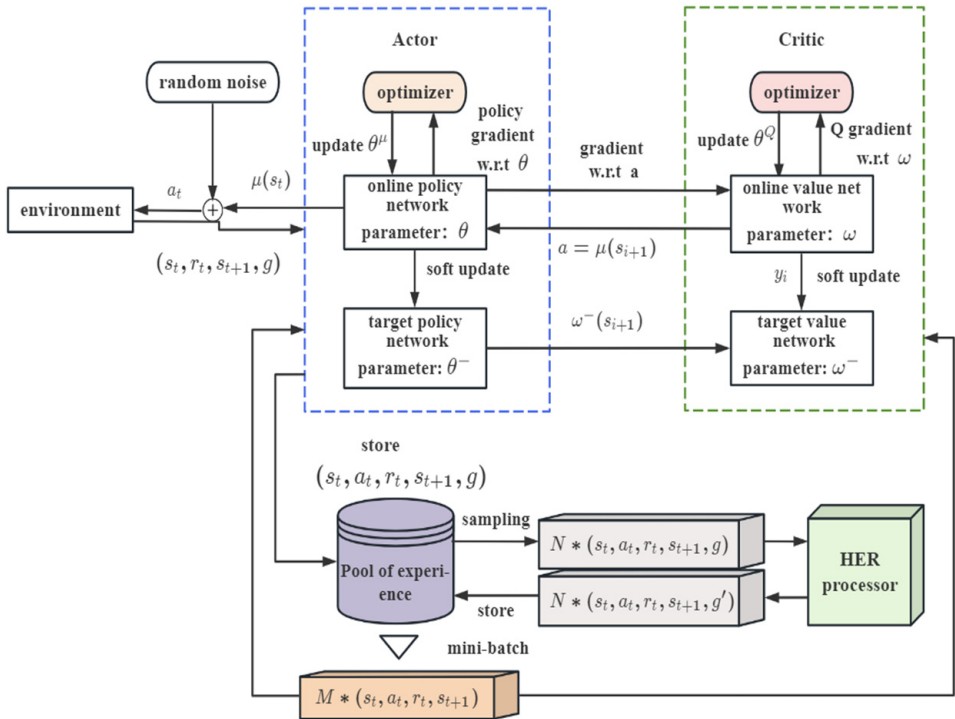

**Figure 4.** HER-DDPG algorithmic network architecture.

### 3.2.2. AUV Dynamic Obstacle Avoidance Mechanism

HER mainly focuses on goal-oriented tasks, and its obstacle avoidance ability for dynamic obstacles may be insufficient. In order to enhance the AUV's capability to handle dynamic environments, specifically to improve its obstacle avoidance ability for dynamic obstacles, the idea of a virtual potential field is introduced to establish the obstacle avoidance function. Based on the obstacle avoidance function, a dynamic obstacle avoidance mechanism for AUVs is designed.

The expression for the repulsive potential field function in a virtual potential field is usually as follows:

$$U_{rep} = \begin{cases} \frac{1}{2}\lambda\left(\frac{1}{\rho} - \frac{1}{\rho^*}\right)^2, & \rho \leq \rho^* \\ 0, & \rho > \rho^* \end{cases} \tag{20}$$

In the given context, $U_{rep}$ represents the repulsive potential field that represents the threat posed by the obstacle. $\lambda$ denotes the constant coefficient of repulsion. $\rho$ represents the Euclidean distance between the robot's current position coordinates and the center of the repulsive potential field region. $\rho^*$ represents the maximum threat distance of the obstacle.

Regarding the process of AUV obstacle avoidance, this study introduces a novel approach that differs from the traditional virtual potential field method. Instead of designing a repulsive potential field centered on obstacles, the approach focuses on designing an AUV repulsive potential field protection domain centered on the AUV itself, thereby establishing the obstacle avoidance function. The schematic diagram provided below illustrates the repulsive potential field protection domain, with the AUV serving as the central reference point (Figure 5).

Considering the uncertainty of dynamic obstacle shapes, this study adopts the initial step of expanding the AUV to create a designated safety space, which is mathematically expressed in Equation (21).

$$\begin{cases} L_1 = \delta \cdot B \\ L_2 = \delta \cdot L \end{cases} \tag{21}$$

In the given context, the expansion coefficient $\delta$ is a constant that has a value greater than 1. Additionally, the variables $L$ and $B$ correspond to the length and width of the AUV, respectively.

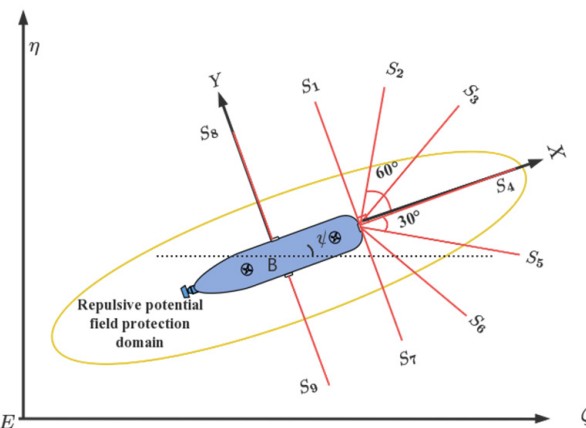

**Figure 5.** AUV repulsive potential field protection domain.

Based on the inflated processed AUV, the idea of repulsive potential field function in the virtual potential field is introduced to establish the obstacle avoidance function of the AUV, and the obstacle avoidance function of the AUV established in this study is shown in Equation (22):

$$U_2(x_t, y_t) = \begin{cases} -k_2\left(\frac{1}{d(x_t, x_0)} - \frac{1}{d_0}\right), & if\,(x_t', y_t') \in \frac{(x_t - x_t')^2}{L_1^2} + \frac{(y_t - y_t')^2}{L_2^2} \leq 1 \\ 0, & if\,(x_t', y_t') \notin \frac{(x_t - x_t')^2}{L_1^2} + \frac{(y_t - y_t')^2}{L_2^2} \leq 1 \end{cases} \tag{22}$$

In Equation (22), $k_2$ represents the gain coefficient. $(x_t, y_t)$ represents the AUV position coordinates at time $t$, while $(x_t', y_t')$ represents the position coordinates of the dynamic obstacle at time $t$. $L_1$ and $L_2$ denote the length and width of the expanded AUV. $d(x_t, x_0)$ represents the distance between the underwater autonomous vehicle and the dynamic obstacle at time $t$. Furthermore, $d_0 = \delta \cdot Max(L_1, L_2)$, which corresponds to the maximum distance influenced by the protection field of the repulsive potential field of the underwater autonomous vehicle.

The obstacle avoidance mechanism of the AUV can be described as follows: Firstly, the AUV travels towards the target along the planned route. When a static or dynamic obstacle enters the protection domain of the AUV's repulsive potential field, the obstacle avoidance task is activated. The AUV experiences a repulsive force based on the obstacle avoidance function, with the magnitude of the force increasing as the distance between the AUV and the obstacle decreases. The AUV continuously adjusts its heading angle. This adjustment continues until the obstacle exits the protection domain of the AUV's repulsive potential field, resulting in a repulsive force of zero on the underwater autonomous vehicle. Once the obstacle is outside the protective field of the AUV's repulsive potential field, the AUV resumes its travel towards the target until it reaches the endpoint.

### 3.2.3. State Space and Action Space of AUV

Moving forward, the state space and action space of the AUV are determined. Acknowledging that a highly intricate and demanding end-to-end motion planning system directly producing AUV thrust mapping is not feasible to learn, this paper adopts a two-layer deep reinforcement learning-based end-to-end approach that integrates perception, planning, and execution. The objectives during the planning stage encompass establishing the state set based on environmental information and defining the action set according to the AUV's motion characteristics.

The motion planning process of AUV is a complex multi-constraint problem whose basic task is to avoid obstacles while reaching the goal point. During the actual movement, the sensor is required to transmit information about the environment and its own state to the AUV, which then outputs a planning policy. The state space $s_t$ can be represented as $s_t = (x_t, o_t)$, where $x_t$ denotes the location information of the target point and AUV. $o_t$ denotes the obstacle information detected by the obstacle avoidance sonar. AUV obtains obstacle information through obstacle avoidance sonar and its own state and target position information through other sensors. The state space during AUV planning is shown in the following Table 2:

**Table 2.** The state space of the planning process.

| State | Symbolic Meaning | Numerical Ranges |
|:---:|:---:|:---:|
| $x_t$ | Location information for the target point and AUV | $x \leq X, y \leq Y, (X, Y)$ is the map boundary |
| $o_t{}^i$ | The distance of the obstacle detected by the obstacle avoidance sonar $i$ | $o_i \in [0, 120], i = 1, \cdots 9$ |

Since in this paper we only study the horizontal plane motion of the underactuated AUV, the action space of the AUV consists of the output action information $a_t$, $a_t = (V_t, \psi_t) \in A$, where $V_t$ denotes the longitudinal velocity of the AUV and $\psi_t$ denotes the heading angle of the AUV. Considering the kinematic and dynamic restrictions of the AUV, the action space model of the AUV is set as follows (Table 3):

**Table 3.** The action space of the planning process.

| Action | Symbolic Meaning | Numerical Ranges |
|:---:|:---:|:---:|
| $V_t$, | Velocity | $V_t \in [-2\,\text{m/s}, 2\,\text{m/s}]$ |
| $\psi_t$ | Heading Angle | $\psi_t \in [-90°, 90°]$ |

### 3.2.4. Reward Function

The quality of the training results and the rate and degree of convergence of the deep reinforcement learning algorithm are directly influenced by the reward function configuration. The reward function of reinforcement learning methods based on hindsight experience replay is usually the most basic binary reward, and the reward of intermediate states is not considered. Therefore, in this paper, we use a combination of HER and reward shaping to train the feedback of AUV, where the reward function considering the intermediate states is built according to the task requirements and constrained model. To achieve global path tracking and obstacle avoidance is the main goal of this study's reward function configuration. The reward value function is created using the established constraint model in the manner described below:

The local desired path of the AUV is to navigate with the global path node as the primary path goal point, avoid obstacles in time, and return to the global route in time when the obstacles are removed, until reaching the goal point, which completes the planning task. In the process of path tracking, the global path is discretized into an ordered sequence of virtual subgoals $\{g_0, g_1, \cdots, g_n\}$. During navigation, the generation of the virtual subgoals sequence takes the form of continuous triggering, that is, when the AUV reaches the $i$th virtual subgoal $g_i$, the previous virtual subgoals are updated and die out. When all virtual subgoal points are extinguished, the goal point is the end of the global path goal.

The reward function of the AUV during training to track the global path is as follows:

$$r_{following} = \begin{cases} -0.5 & Away\ from\ the\ goal \\ +0.5 & Close\ to\ the\ goal \\ 4.0 & achieve\ goal \\ 0 & other\ conditions \end{cases} \tag{23}$$

In particular, the goal can be either the end point of the global path target $g$, or a virtual subtarget $g_i$. $d_g = d_{t+1} - d_t$ determines how close or far the AUV is from the target location, and $d_g$ represents the difference between those distances at two different times (times $t + 1$ and $t$). Considering the constraint, when $L_{target} \leq 2$ m, the AUV is considered to reach the goal.

The reward function of the obstacle avoidance module is designed based on the obstacle avoidance mechanism proposed in Section 3.2.2. The objective of obstacle avoidance is to allow the AUV to effectively avoid unknown obstacles. In this paper, we design an intermediate reward based on the obstacle avoidance function of the AUV, in addition to the traditional discrete negative reward for collision with an obstacle. That is, when an obstacle detected by the AUV's sonar does not enter the domain of the AUV's repulsive potential field, the AUV is safe, and the reward value is 0. The AUV receives a continuous negative reward when a static or dynamic obstacle enters the domain of the AUV's repulsive potential field. This reward value is provided by the obstacle avoidance function designed in the previous section, which states that the closer the obstacle is to the AUV, the more negative the reward the AUV receives. If the dynamic obstacle eventually reaches within the safe radius of the AUV for obstacle avoidance, the two collide and receive a negative reward value of $-1.0$. The specific reward function of the obstacle avoidance module is set as follows:

$$r_{avoid} = \begin{cases} -k\left(\frac{1}{d(x_t, x_0)} - \frac{1}{\delta Max(L_1, L_2)}\right) & if (x_t', y_t') \in \frac{(x_t - x_t')^2}{L_1^2} + \frac{(y_t - y_t')^2}{L_2^2} \leq 1 \\ 0 & if (x_t', y_t') \notin \frac{(x_t - x_t')^2}{L_1^2} + \frac{(y_t - y_t')^2}{L_2^2} \leq 1 \\ -1.0 & if (x_t', y_t') \in (x_t - x_t')^2 + (y_t - y_t')^2 \leq d_s^2 \end{cases} \tag{24}$$

In the given context, $k$ represents the gain coefficient, which is set to 0.1. The expansion coefficient $\delta$ is a constant that takes a value greater than 1, and in this paper, it is specifically set as $\delta = 3.0$. $d_s$ represents the minimum safe distance of the AUV. Based on the constraints established in the previous section, the safe radius for obstacle avoidance of the AUV is defined as 5 m, which means $d_s = 5$ m.

Considering the navigation path length constraint, the reward value function for AUV roaming is as follows:

$$r_{navigation} = -0.05 \tag{25}$$

This means that for each action performed by the AUV, a reward of $-0.05$ is obtained. Thus, the total reward function is a weighted sum of three terms:

$$r = k_1 r_{following} + k_2 r_{avoid} + k_3 r_{navigation} \tag{26}$$

where $k_1$, $k_2$, and $k_3$ denote the scaling coefficients of the three reward modules, respectively.

The procedure to obtain the reward function when the AUV performs planning training is as Figure 6:

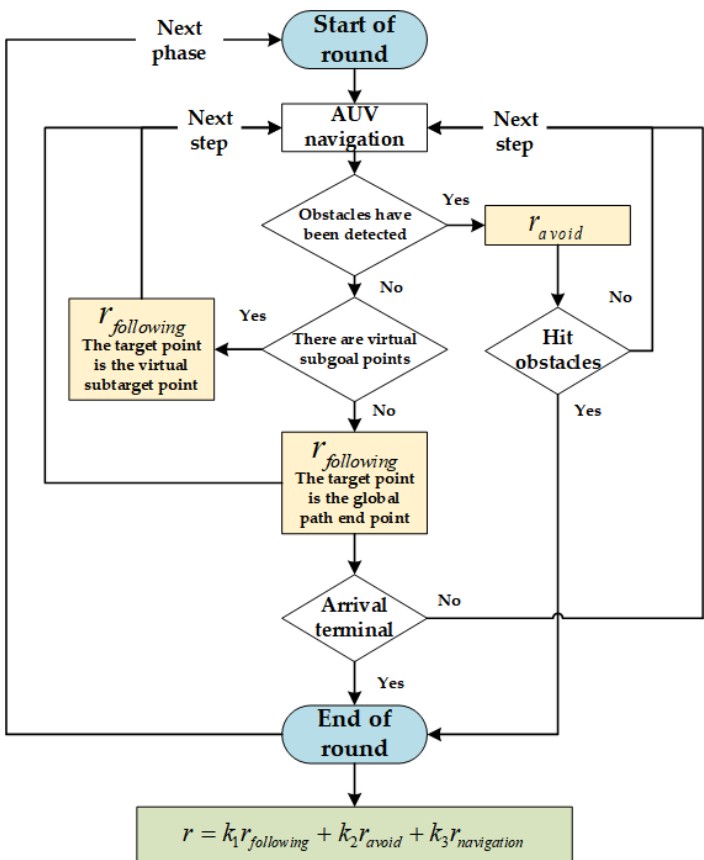

**Figure 6.** Flowchart of the reward function acquisition.

*3.3. Execution Method Based on the DDPG Algorithm*

3.3.1. State Space and Action Space of AUV

Once the action planning phase is completed, the system proceeds to the control layer, where the corresponding control force needs to be assigned to guide the AUV in executing the planned action. To achieve this, it becomes essential to establish the executive force set. Additionally, a mapping relationship is established from the action set to the executive force set through the movements determined during the planning phase. This mapping plays a crucial role in implementing the control aspect of motion planning. The underlying idea behind constructing the mapping relationship between the action set and the execution set is that the target action derived from the planning layer corresponds to the thrust and torque required for the AUV. As a result, deep reinforcement learning algorithms are trained to facilitate the mapping between the motion state and the applied forces.

The state space of the control layer is the action $a$ output from the planning layer, and the action space is the force $\tau$ of the control system. Combined with the aforementioned Markov decision process, the design flow of the execution method based on the DDPG algorithm can be briefly described as follows: Initially, the actions obtained through planning training for the AUV are employed as inputs to the control system. Drawing from the experience gained during training, the DDPG algorithm controller maps the target commands to the necessary thrust and torque for AUV motion. These control forces are then applied within the AUV's dynamic model, resulting in the output of actual motion parameters corresponding to the current control force. The feedback loop encompasses comparing the actual motion parameters with the planned target commands and utilizing the discrepancy as feedback to the DDPG control system. Concurrently, a reward function is defined based on the control discrepancy, aiming to minimize the control difference of the DDPG control system towards zero. This iterative process facilitates continuous updates, enabling the effective execution of the desired action.

The AUV's state in the control layer consists mainly of the velocity and heading angle of the AUV planning layer:

$$s = [V, \psi] \tag{27}$$

where $V$ represents the AUV navigation velocity, and $\psi$ represents the AUV heading angle.

According to the DDPG policy formulation, the action output is expressed as a function as follows:

$$\tau = a = \mu(s_t | \theta^\mu) \tag{28}$$

As a result, the control layer's output action can be stated as follows:

$$\tau = \mu(s_t) = \mu(V(t), \psi(t)) \tag{29}$$

The above equation represents the mapping of the system from state to action, and the output control force is used as input to the dynamical equation to obtain the actual motion parameters. The next is to update the neural network parameters in the algorithm by continuously training.

### 3.3.2. State Space and Action Space of AUV

The ultimate goal of the execution system is to have the actual motion parameters output by the AUV model be identical to the planned target commands, that is, the difference values tend to zero. The reward function of the system, which is based on this control objective, is taken to be the absolute value of the difference between the actual motion state of the AUV and the desired state, i.e.,

$$r_v = -|e_V|, \ r_\psi = -|e_\psi| r = r_v + r_\psi \tag{30}$$

where $e_V$ and $e_\psi$ represent, respectively, the discrepancy between the anticipated target commands and the actual target directives for velocity and heading.

According to the reward function, the action–execution method based on the DDPG algorithm is designed as in Figure 7:

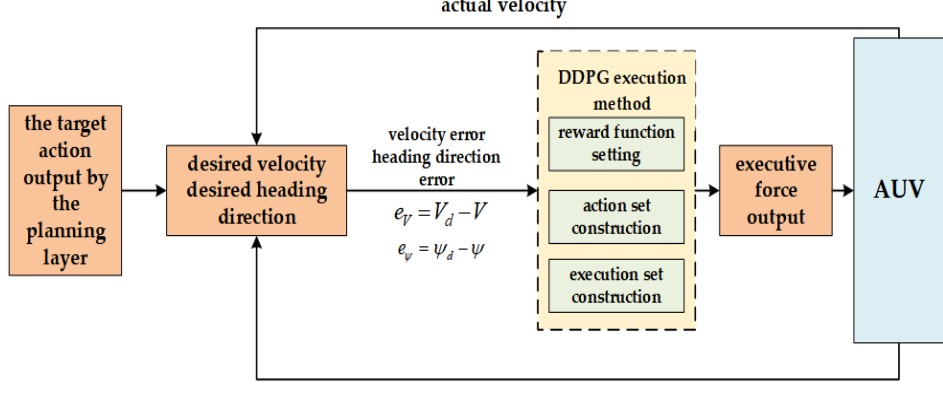

**Figure 7.** Action–execution mapping process.

### 4. Simulations

In this section, simulation tests are conducted to assess the effectiveness of the proposed end-to-end AUV motion planning system, which is based on the two-layer deep reinforcement learning algorithm. The tests employ the suggested AUV model and algorithm to evaluate the system's performance. Initially, the planning method and execution method are validated independently. Subsequently, a local motion planning synthesis experiment is carried out using a two-layer end-to-end system, with a complex tunnel environment serving as the typical task background. For this experiment, the neural network is

constructed using Pytorch, visual simulation is managed using the Python Pyglet package, and the model is trained using a GPU.

The neural networks employed in this study exhibit a limited number of layers, and the output of these networks necessitates clipping. Therefore, the activation function chosen for these networks is tanh, which represents the hyperbolic tangent function.

$$\tanh(x) = \frac{e^x - e^{-x}}{e^x + e^{-x}} \tag{31}$$

When updating the parameters of the neural network, the artificial neural network is trained by using the error backpropagation algorithm combined with gradient descent. Neural networks in deep reinforcement learning do not require too many layers for planning problems. In general, 2–3 layers are chosen, as too large a network structure leads to parameter redundancy and overfitting. The neural network architecture with two hidden layers is chosen according to the dimensionality of the AUV state space and action space set in this study. The hidden layers are fully connected layers, each of which contains 256 neurons.

The parameter settings for the learning phase of the deep reinforcement learning algorithm are provided in the following Table 4:

**Table 4.** DDPG and HER-DDPG algorithm training parameters.

| Parameter | Value |
| --- | --- |
| Learning rate for actor $\alpha_1$ | 0.001 |
| Learning rate for critic $\alpha_2$ | 0.001 |
| Discount factor $\gamma$ | 0.9 |
| Initial exploration $\varepsilon_0$ | 1 |
| Final exploration $\varepsilon_{end}$ | 0.01 |
| Neural network hidden layer | 2 |
| Number of neurons in each hidden layer | 256 |
| Replay memory size | 50,000 |
| Minibatch size $m$ | 64 |
| Soft-update frequency $\tau$ | 0.01 |
| Final exploration frame $T$ | 2000 |
| Max episodes $M$ | 5000 |

The hardware configuration environment of the motion planning computer on which the simulation experiments were performed is detailed in the following Table 5:

**Table 5.** Motion planning computer hardware configuration.

| Name | Configuration |
| --- | --- |
| Computer model | X64 compatible with desktop computers |
| Operating system | Windows 10 Enterprise 64 bit |
| Processor | Intel Core i9-7980 XE @ 2.60 GHz |
| Mainboard | ASUS PRIME X299-A |
| Internal storage | 32 GB |
| Primary hard drive | Samsung NVMe SSD 970 (512 GB) |
| Graphics card | Nvidia TITIAN V (12 G) |

### 4.1. Perception–Planning Experiment

First, the HER-DDPG planning method designed in this paper is used to perform perception–planning experiments, namely mapping experiments from environmental information to AUV movement. In this paper, we construct a large number of environmental maps for various tasks such as target orientation and obstacle avoidance and conduct a series of simulation experiments to test the effect of the designed algorithm.

Pretraining is needed before testing the planning results of the algorithm. In this paper, the DDPG and HER-DDPG algorithms are pretrained, and the improved HER-DDPG

algorithm is compared with the results of the ordinary DDPG training process. Figure 8 shows the cumulative reward values of the DDPG and HER-DDPG algorithms in a single round during training.

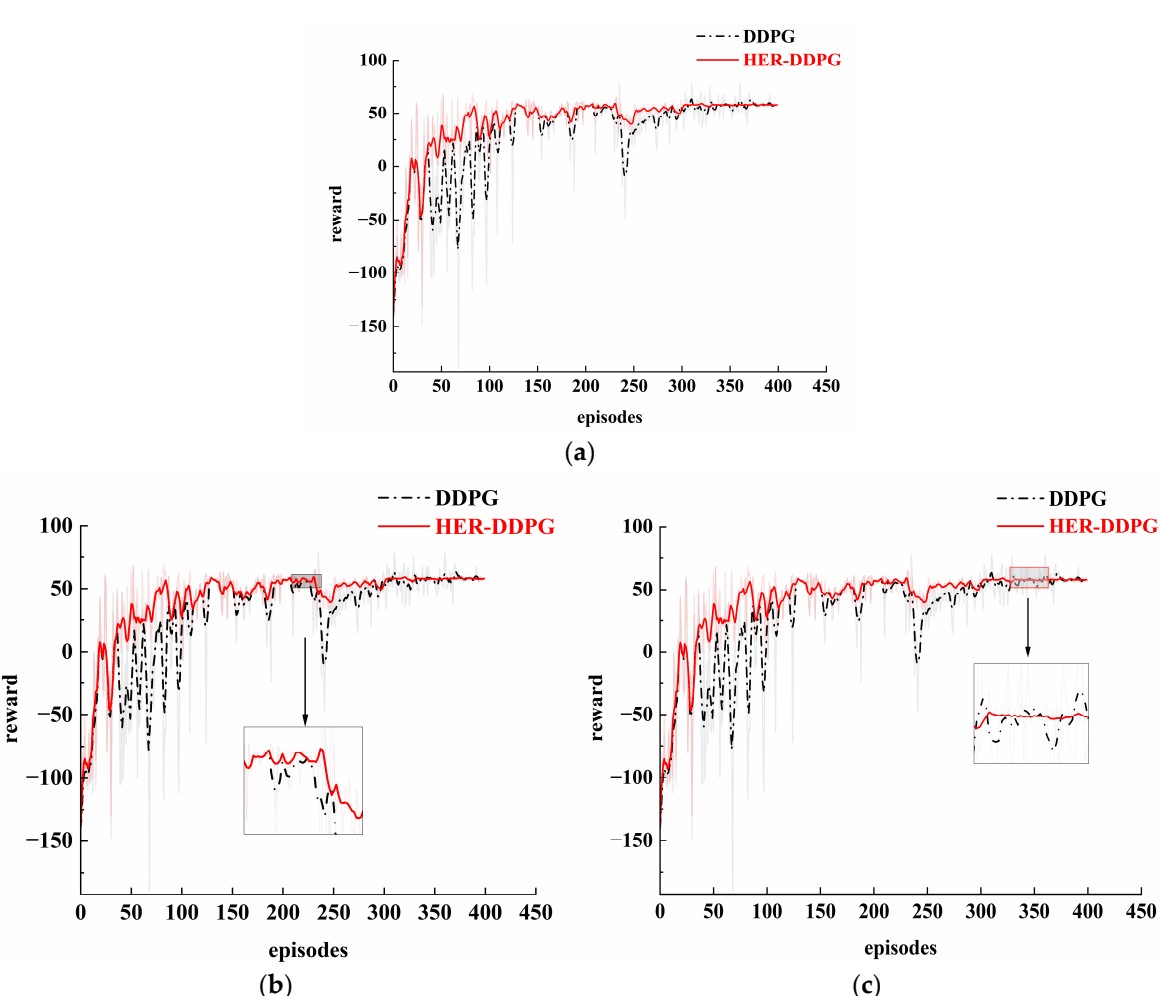

**Figure 8.** Record and compare the reward values: (**a**) total reward comparison diagram; (**b**) intermediate stage; (**c**) end stage.

The number of training episodes is represented along the horizontal axis in the upper panel. A total of 2000 training sessions were completed, with the average reward value recorded every five epochs. As shown in the aforementioned figures, both the HER-DDPG-based method and the standard DDPG algorithm display learning curves that converge to satisfactory values. Notably, the overall reward value of HER-DDPG surpasses that of DDPG. Although DDPG may exhibit higher reward values in some rounds, the learning process based on HER-DDPG showcases enhanced stability and faster convergence.

The trained neural network model is saved once training is finished so that the effectiveness of the planning technique may be evaluated. Figures 9 and 10 show the test results for AUV completion trends for single-target, multitarget, and obstacle avoidance based on the HER-DDPG planning approach. The operating environment of the AUV movement is represented in the figure by the square area with solid black lines, whose length and width are 600 m $\times$ 600 m, the red rectangle serving as the AUV, the black polygon area serving as the obstacle, the yellow circular area serving as the target area, and the blue curve serving as the actual AUV navigation path.

Perception–Planning Test Experiment 1: Goal-directed behavior experiments

AUV goal-directed behavior is essential for accomplishing various navigation and exploration tasks. This section conducts goal-directed behavior experiments to assess the ef-

fectiveness of the action planning algorithm. The experiments are divided into four groups, encompassing not only straightforward scenarios like single-target and multitarget simulations but also more complex real-world tasks. These tasks include wide-field water search and river waterway search traversal, designed to replicate challenging scenes encountered in practical situations. In the search task, the AUV follows upper-level instructions to sequentially navigate to target points along a predetermined path, utilizing environmental information. The ultimate objective is to reach the end of the path and successfully complete the exhaustive search task. The parameters for each set of experiments are provided below (Table 6).

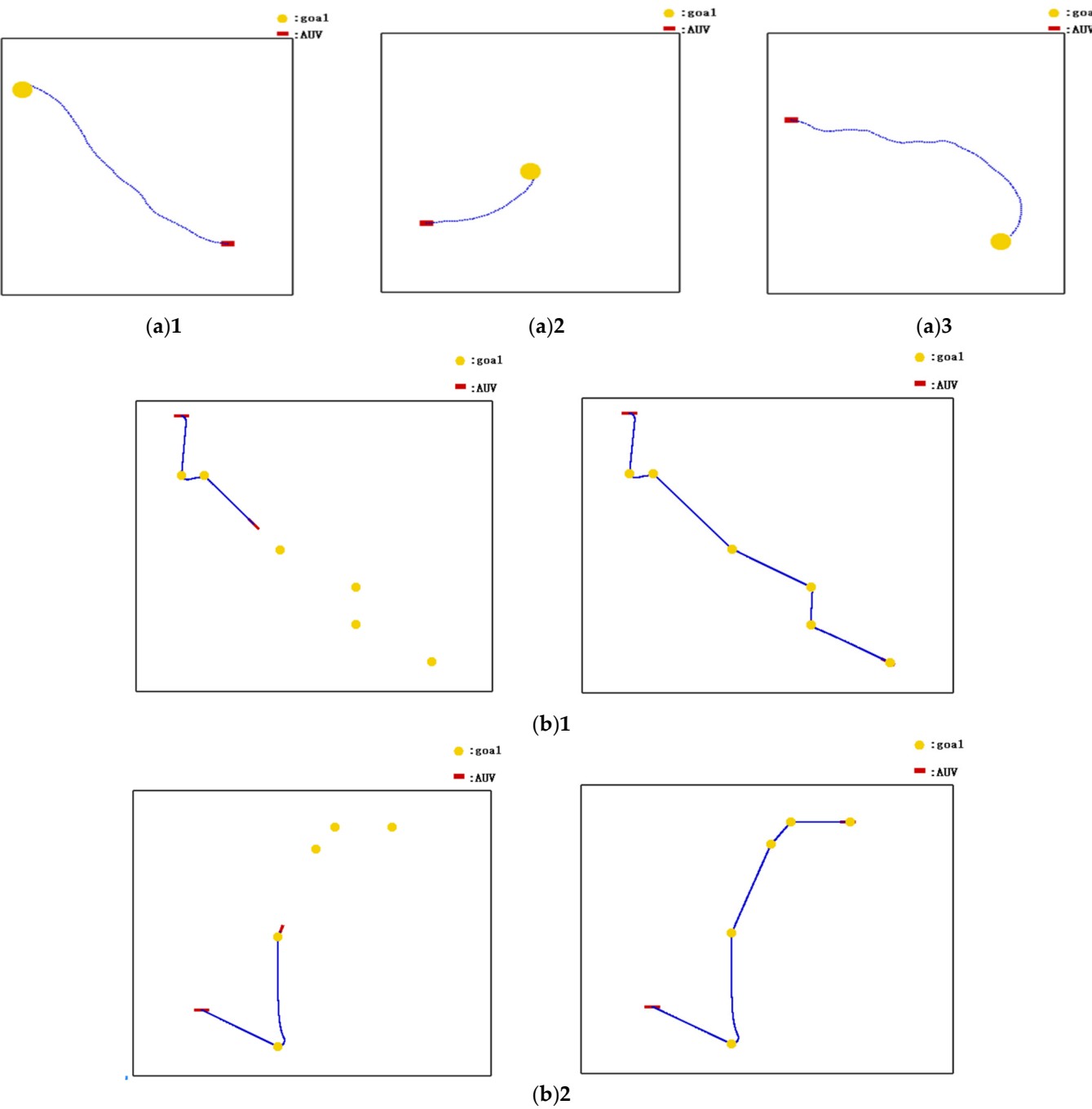

**Figure 9.** *Cont.*

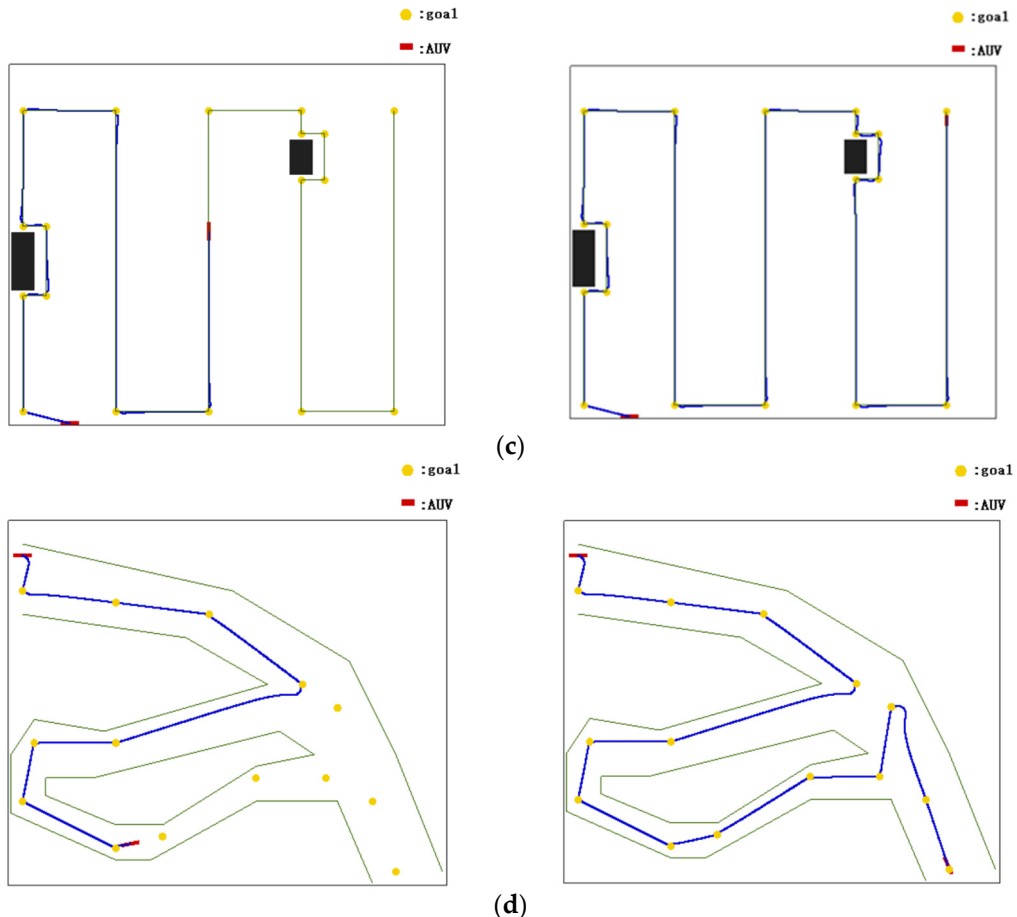

**Figure 9.** Experimental results of the goal-directed behavior simulation experiment based on the HER-DDPG planning algorithm: (**a**) test results from the single target experiments; (**b**) test results from the multitarget experiments; (**c**) test results from the wide-field water search experiment; (**d**) test results from the river waterway search traversal experiment.

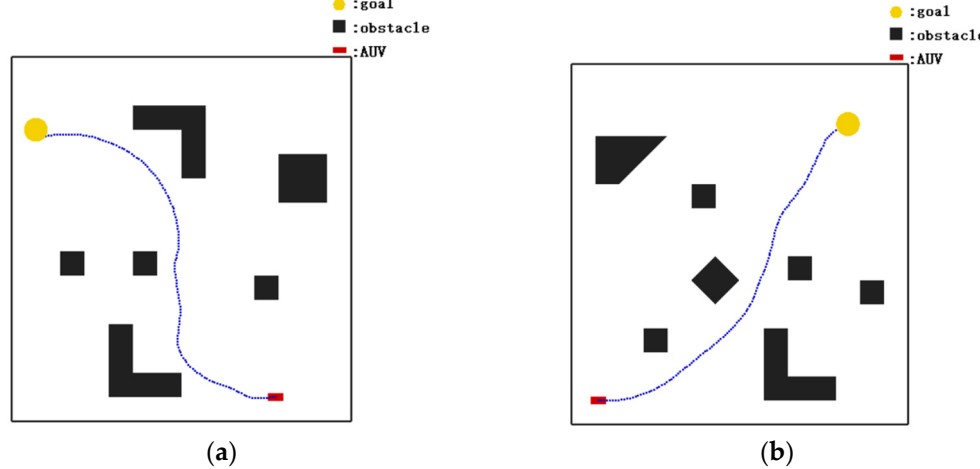

**Figure 10.** Experimental results of AUV static obstacle avoidance based on the HER-DDPG planning algorithm: (**a**) static obstacle avoidance environment a; (**b**) static obstacle avoidance environment b.

**Table 6.** Environmental parameters for the goal-directed behavior simulation experiments.

| Experimental Serial Number | Starting Position of the AUV (m) | End-Target Location (m) | Target Area Diameter (m) | Initial Velocity (m/s) | Target Velocity (m/s) |
|---|---|---|---|---|---|
| (a)1 | (300, 100) | (50, 325) | 25 | 0 | 1.0 |
| (a)2 | (75, 125) | (200, 200) | 25 | 0 | 1.0 |
| (a)3 | (50, 275) | (300, 100) | 25 | 0 | 1.0 |
| (b)1 | (70, 380) | (400, 50) | 25 | 0 | 1.0 |
| (b)2 | (100, 100) | (425, 75) | 25 | 0 | 1.0 |
| (c) | (75, 25) | (425, 350) | 25 | 0 | 1.0 |
| (d) | (25, 375) | (425, 25) | 25 | 0 | 1.0 |

The test results are shown below:

The test results demonstrate that once the AUV initiates its journey from the starting region, the HER-DDPG algorithm promptly adjusts the target heading and desired velocity to ensure smooth navigation towards the target region. Remarkably, this adjustment is effective even when the target region appears randomly within the AUV's operational environment. In the case of multitarget tasks, the HER-DDPG algorithm dynamically plans the AUV's movement in real-time, facilitating the AUV reaching each target point sequentially. Similarly, in the search scenarios, the HER-DDPG algorithm guides the AUV to systematically approach each target point along the designated path, ultimately accomplishing the search traversal task by reaching the endpoint. Visual observations from Figure 9a–d provide compelling evidence that, given the applicable constraint requirements, the HER-DDPG algorithm generates routes that closely approximate optimality without including any invalid routes.

Perception–Planning Test Experiment 2: Obstacle avoidance in unknown environments.

To assess the effectiveness of the horizontal plane HER-DDPG planning method and the dynamic obstacle avoidance mechanism proposed in this paper for AUV obstacle avoidance in unknown environments, a series of avoidance experiments are designed. These experiments encompass scenarios with both static and dynamic, unknown obstacles.

Firstly, the static obstacle avoidance experiments were conducted. In these experiments, the AUV obstacle avoidance planning system operates in an environment where the information regarding the obstacles, such as their number and positions, is unknown. To detect static obstacles, the AUV relies on its obstacle avoidance sonar and other onboard sensors. The specific parameters of the simulation environment used in the static obstacle avoidance experiment are detailed in the following Table 7.

**Table 7.** Environmental parameters for the static obstacle avoidance simulation experiments.

| Experimental Serial Number | Starting Position of the AUV (m) | Target Location (m) | Target Area Diameter (m) | Number of Obstacles | Safe Distance from Obstacles (m) | Initial Velocity (m/s) | Target Velocity (m/s) |
|---|---|---|---|---|---|---|---|
| a | (300, 50) | (50, 325) | 25 | 6 | $\geq 5$ m | 0 | 1.0 |
| b | (50, 50) | (325, 325) | 25 | 7 | $\geq 5$ m | 0 | 1.0 |

The test results of the static obstacle avoidance experiments are as follows:

From the above results, it is evident that the historical trajectories of the AUVs (represented by the blue dashed line in the figure) depict their inclination to navigate towards the target, guided by the effective HER-DDPG planning method, in the absence of any detected obstacles. Conversely, when the AUVs identify obstacles, they employ the HER-DDPG planning algorithm to strategize obstacle avoidance maneuvers, ensuring successful circumvention of obstacles and ultimately reaching the target area in a safe manner.

Next, dynamic obstacle avoidance experiments were conducted. In these experiments, the AUV follows a predetermined movement route while multiple unknown dynamic

obstacles are introduced simultaneously. Each of these obstacles adheres to a pre-designed path. As the moving obstacles enter the detection range of the AUV's obstacle avoidance sonars, the AUV is able to acquire information about the obstacles. The specific parameters of the simulation environment for the dynamic obstacle avoidance experiment are detailed in the following Table 8.

**Table 8.** Environmental parameters for the dynamic obstacle avoidance simulation experiments.

| Experimental Serial Number | Starting Position of the AUV (m) | Destination (m) | Velocity of Unknown Obstacles (m/s) | Number of Unknown Obstacles | Safe Distance from Obstacles (m) | Initial Velocity (m/s) | Target Velocity (m/s) |
|---|---|---|---|---|---|---|---|
| a | (40, 30) | (200, 350) | 0.1 | 2 | ≥5 m | 0 | 1.0 |
| b | (30, 200) | (350, 200) | 0.2 | 4 | ≥5 m | 0 | 1.0 |

The test results of the dynamic obstacle avoidance experiments are as follows (Figure 11):

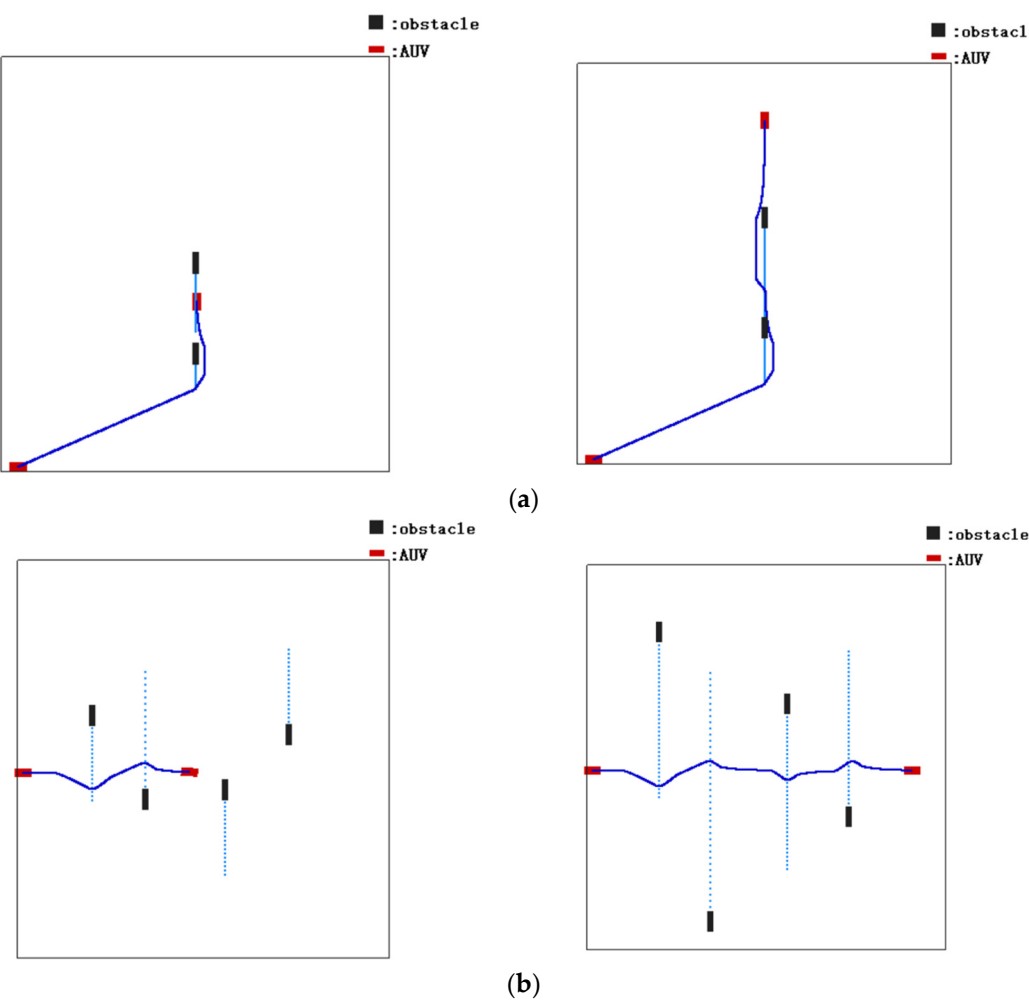

**Figure 11.** Experimental results of AUV dynamic obstacle avoidance experiments based on the HER-DDPG planning algorithm: (**a**) horizontal dynamic obstacle avoidance; (**b**) vertical dynamic obstacle avoidance.

Based on the above results, it can be observed that the HER-DDPG algorithm effectively guides the AUV along a predetermined route when the dynamic obstacle remains outside the AUV's repulsive protection domain. However, upon the dynamic obstacle entering the AUV's repulsive protection domain, the AUV triggers the dynamic obstacle

avoidance mechanism. Through the combined guidance of the HER-DDPG algorithm and the dynamic obstacle avoidance mechanism, the AUV dynamically adjusts its bow angle and speed to successfully circumvent the dynamic obstacle. Once the avoidance is accomplished, the AUV seamlessly returns to its original route. These experiments substantiate the AUV's dynamic responsiveness and its ability to navigate dynamic environments using the planning algorithm proposed in this study.

In conclusion, the simulation results demonstrate that the planning strategy developed in this paper may successfully invoke the deep reinforcement learning network to efficiently direct the AUV to the goal in the absence of obstacle detection. When the threat of obstacles is detected, the obstacle is avoided in time with real-time dynamic planning. After reaching a safe area, the heading direction was adjusted to continue the target task. Therefore, the proposed perception–planning method is feasible and effective for AUV motion problems in unknown environments, and the planning network has strong generalization ability to adapt to different environments.

### 4.2. Planning–Execution Experiment

Next, the planning–execution experiment is conducted, serving as a mapping experiment that connects the planning information obtained from the AUV to the execution of the AUV's propulsion. This experiment also functions as an AUV motion control experiment. Based on the mapping strategy acquired from the training conducted in this paper, a comparison is made between the DDPG algorithm and the S-plane control algorithm. Furthermore, separate experiments are performed to assess the AUV's ability to follow straight paths and curves, respectively. The experimental errors are recorded to validate the algorithm's reliability.

Planning–Execution Test Experiment 2: Comparison of DDPG control algorithm and S-plane control algorithm

S-plane control learns from the PID method based on fuzzy control [26]. The algorithm is expressed as follows:

$$
\begin{cases}
u_i = 2/(1 + \exp(-k_{i1}e_i - k_{i2}\dot{e}_i)) - 1 + \Delta u_i \\
f_i = K_i u_i
\end{cases}
\tag{32}
$$

The S-plane control algorithm and the proposed DDPG control algorithm are used to control the AUV to sail at a target speed of 1.0m/s and a target heading of $\pi/12$. In order to compare the anti-interference ability of the algorithm, instantaneous interference, constant interference, and periodic interference were applied to the AUV in different time periods of navigation. The results of the simulation test are shown in Figure 12.

In the simulation, a burst force of 180 N is applied laterally for 10 steps while the AUV is traveling for 250 steps. A constant force of 30 N in the $E - \eta$ direction in the geodesic coordinate system is applied to the AUV for 350 steps while it is sailing. As the AUV sails for 750 steps, a periodic perturbation force of size $25 \cdot \sin(0.01\pi t)$ N in the $E - \eta$ direction in the geodesic coordinate system is applied to it for 350 steps.

The mean absolute error (MAE) comparing the two algorithms for controlling AUV velocity and heading is shown in Table 9.

**Table 9.** Comparison of the mean absolute error of the control.

| | Instantaneous Interference | | Constant Interference | | Periodic Interference | |
|---|---|---|---|---|---|---|
| | MAE-v (m/s) | MAE-Theta (rad) | MAE-v (m/s) | MAE-Theta (rad) | MAE-v (m/s) | MAE-Theta (rad) |
| S-plane | 0.0080 | 0.0089 | 0.0460 | 0.0028 | 0.0421 | 0.0448 |
| DDPG | 0.0032 | 0.0041 | 0.0174 | 0.0001 | 0.0045 | 0.0030 |

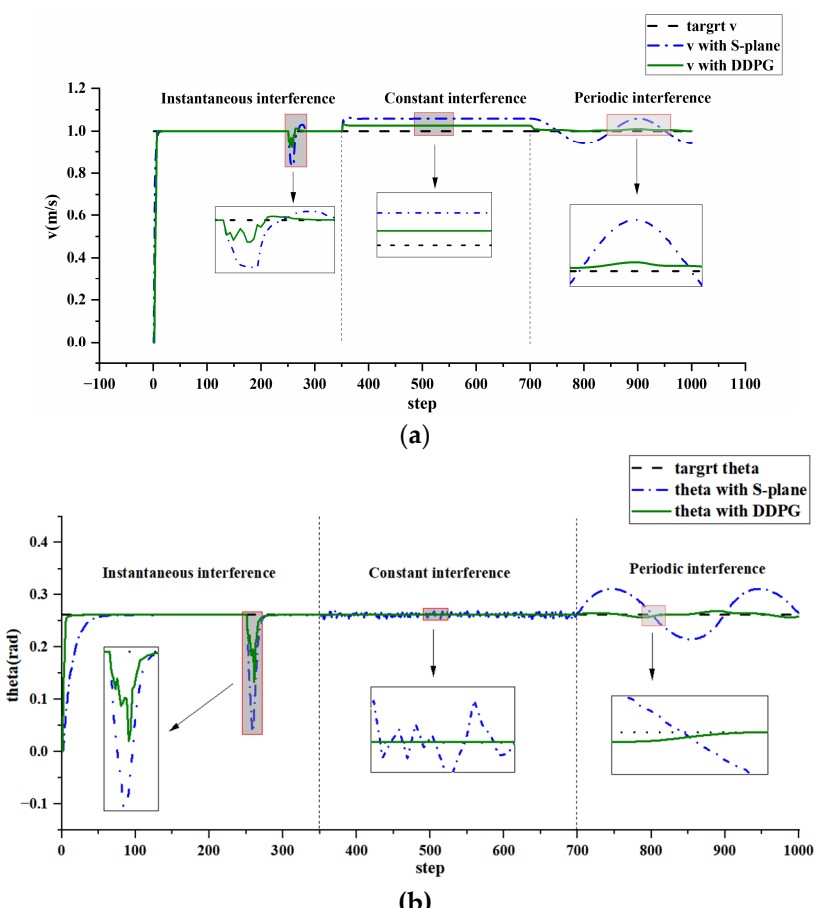

**Figure 12.** Comparison of S-plane and DDPG control methods (The red boxes are intercepts.): (**a**) result of velocity control; (**b**) result of heading control.

From the test results, it can be seen that the velocity and heading of the AUV under the two algorithms are greatly deviated when the instantaneous force is applied, but after the force disappears, the DDPG algorithm can control the AUV to the target command as much as possible, and the control error is smaller than that of the S-plane algorithm. Both the S-plane algorithm and the DDPG algorithm have some bias in controlling the AUV velocity under constant force interference, but as shown in Table 9, the mean absolute error of the DDPG in controlling the AUV velocity is significantly smaller than the S-plane result. For heading tracking, the results controlled using the S-plane algorithm have significant oscillations, while the results controlled using the DDPG algorithm have very small oscillations, which are almost negligible. Under the interference of periodic forces, the DDPG-controlled AUV performs significantly better than the S-plane control, and the AUV can navigate well according to the target commands.

From the above simulation tests, it can be seen that the proposed DDPG based mapping control algorithm achieves significantly better control than the basic S-plane controller under various perturbations without changing the algorithm parameters.

Planning–Execution Test Experiment 2: straight-line path following

In the simulation environment, a linear target path has been defined. Its beginning and finishing points are located at (200, 40) and (300, 300), respectively. The AUV's starting point is at (70, 40), its starting speed is 0 m/s, and its intended speed is 1.0 m/s. A simulation test of the AUV's straight-line path following planning task was conducted using the DDPG-based execution approach trained in this research, and the results are displayed in the Figure 13:

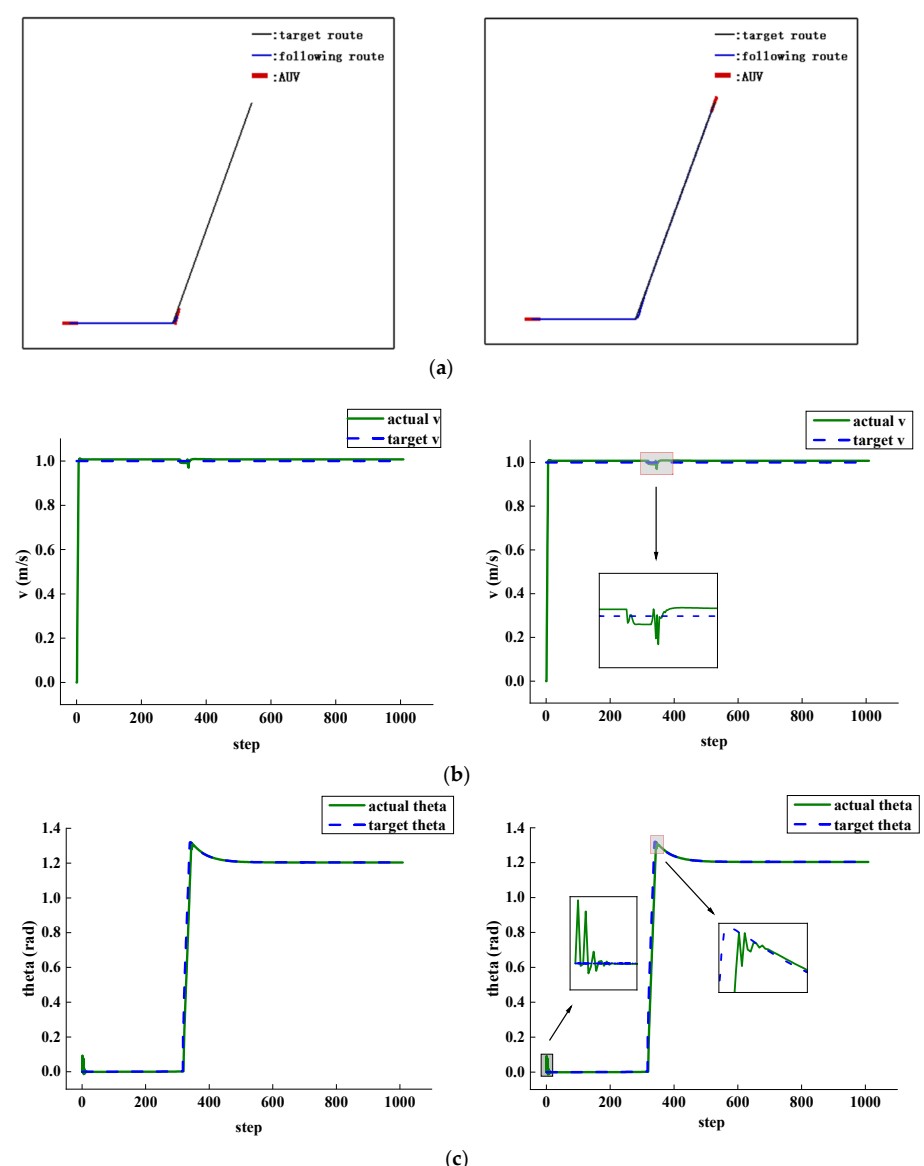

**Figure 13.** AUV straight-line path following experimental result based on DDPG execution algorithm: (**a**) path following result; (**b**) velocity following result; (**c**) heading following result.

The following Table 10 shows the mean absolute error and maximum absolute error of each following variable in the process of AUV following the straight path, removing the large error at the beginning of the stage of driving to the target path.

**Table 10.** The experimentally determined errors of the straight path.

| Error Pattern | Velocity (m/s) | Heading Angle (Rad) |
|---|---|---|
| MAE | 0.00703 | 0.02618 |
| MAX | 0.03034 | 0.04697 |

On the basis of the planning–execution mapping technique presented in this research, it can be shown from the path following findings in Figure 13a that the AUV can follow the global path with ease. Although the following path deviates slightly, the error is virtually zero and there is a fluctuation at the corners, which is consistent with AUV dynamic constraints. Based on the velocity following graph, the heading following graph, and the error table, it can be seen that the AUV can achieve the target velocity and heading direction while following the

path with minor errors based on the designed planning-execution mapping strategy. Based on the steps corresponding to the error fluctuation, it can be seen that the error fluctuation is located at the AUV starting point and the path turning point, which is consistent with the dynamical constraint, and the error quickly goes to zero.

Planning–Execution Test Experiment 3: curve path following

Constructing a curve path in a simulation environment. Function of the curve path:

$$y = 30 \cdot \sin(0.02x) + 180(65 \leq x \leq 500) \tag{33}$$

The AUV's starting point was at (35, 230), its initial speed was 0 m/s, and its intended speed was 1.0 m/s. The curve path following problem of an AUV is simulated using the DDPG-based execution approach trained in this paper, and the results are displayed in the Figure 14:

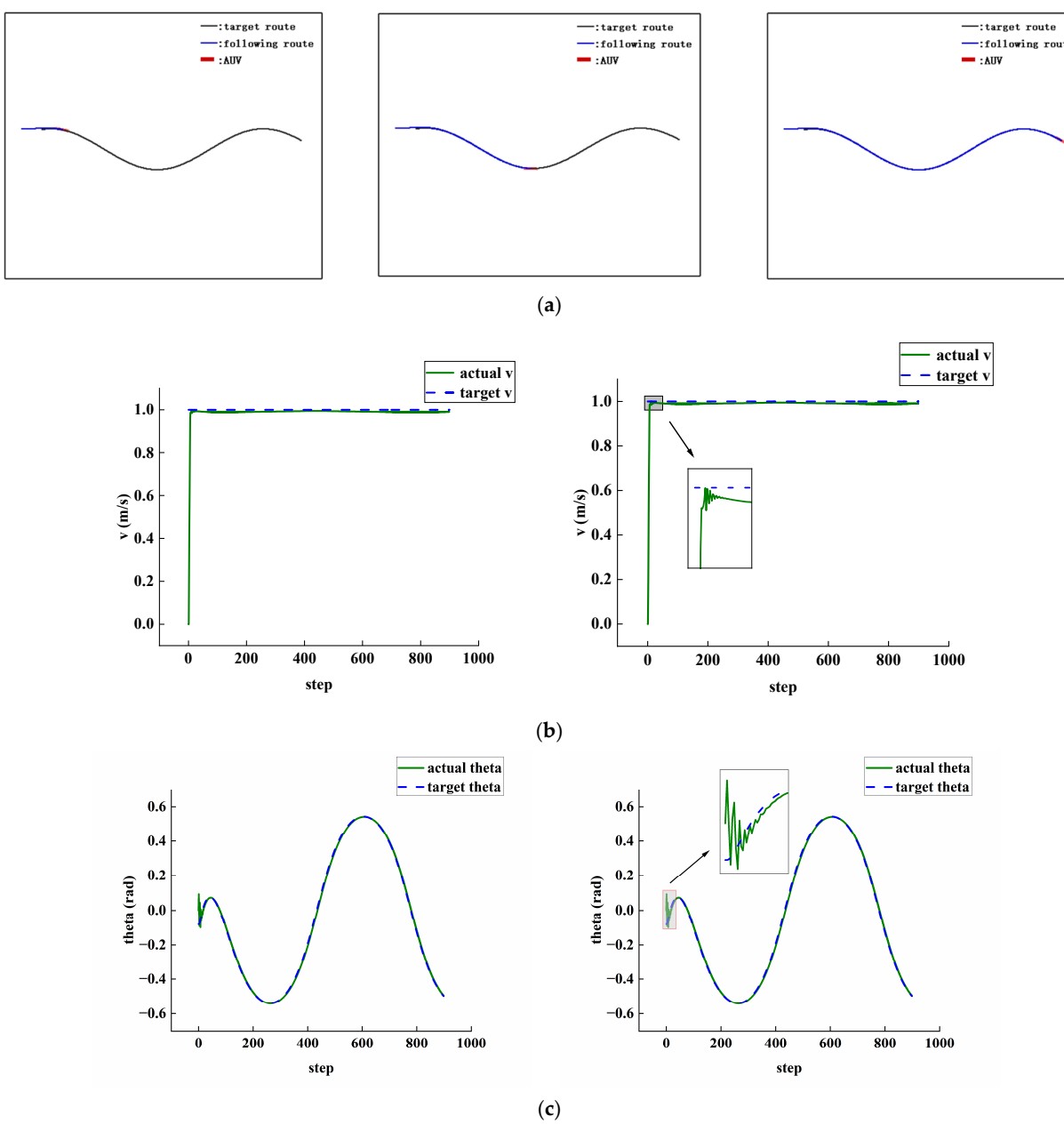

**Figure 14.** AUV curve line path following experimental result based on DDPG execution algorithm: (**a**) path following result; (**b**) velocity following result; (**c**) heading following result.

The following Table 11 displays the average absolute error and maximal absolute error for each subsequent variable in the process of an AUV following a curve path, erasing the significant inaccuracy at the start of the driving stage to the target path.

**Table 11.** The experimentally determined errors of the curve path.

| Error Pattern | Velocity (m/s) | Heading Angle (Rad) |
|---|---|---|
| MAE | 0.01025 | 0.03201 |
| MAX | 0.05763 | 0.07950 |

From the test results and error tables, it is easy to see that, like the results for the following straight line, AUV can nicely follow the curve path based on the execution strategy trained in this paper. The AUV yaws somewhat, and its motion parameters change at the path's beginning and ending points, but it can quickly return to the target instruction and target path. A suitable range encompasses the control mistake.

### 4.3. Comprehensive Experiment

In this section, we combine the perception–planning and planning–execution modules to conduct a comprehensive end-to-end planning experiment based on the proposed architecture of the local motion planning system. The underwater environment, particularly narrow water bodies like underwater tunnels, presents various challenges, including poor visibility, limited navigation range, and numerous wall obstacles. These factors significantly increase the complexity of AUV motion planning. Therefore, we focus on AUV tunneling detection as a representative task scenario, where the known global route is defined by the tunneling central axis, allowing us to design simulation tests accordingly. We implement local motion planning to achieve both global path following and the avoidance of unknown obstacles during AUV tunnel detection. To demonstrate the impact of AUV action planning conveniently and to simplify the tunneling environment and global route, we present the experimental setup in Figure 15.

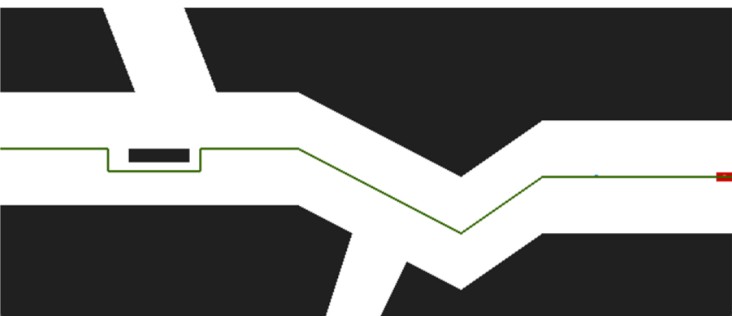

**Figure 15.** Global planning path for tunnel inspection.

The environment parameters are as follows (Table 12):

**Table 12.** Environmental parameters for tunnel planning simulation experiment.

| Starting Position of the AUV (m) | Ending Position of the AUV (m) | Position of the Known Obstacle (m) | Velocity of the Unknown Obstacle (m/s) | Safe Distance from Obstacles (m) | Initial Velocity (m/s) | Target Velocity (m/s) |
|---|---|---|---|---|---|---|
| (670, 125) | (0, 150) | Center coordinate: (150, 140) Length: 28 m; width: 6.5 m | 0.1 | >5 m | 0 | 1.0 |

Sudden unknown obstacles are set on the global planning path, and based on the above designed end-to-end perception–planning–execution approach, the global path tracking and obstacle avoidance of unknown moving obstacles for AUV is achieved by constructing a mapping of state–action–execution. In Figure 16, the test's findings are displayed.

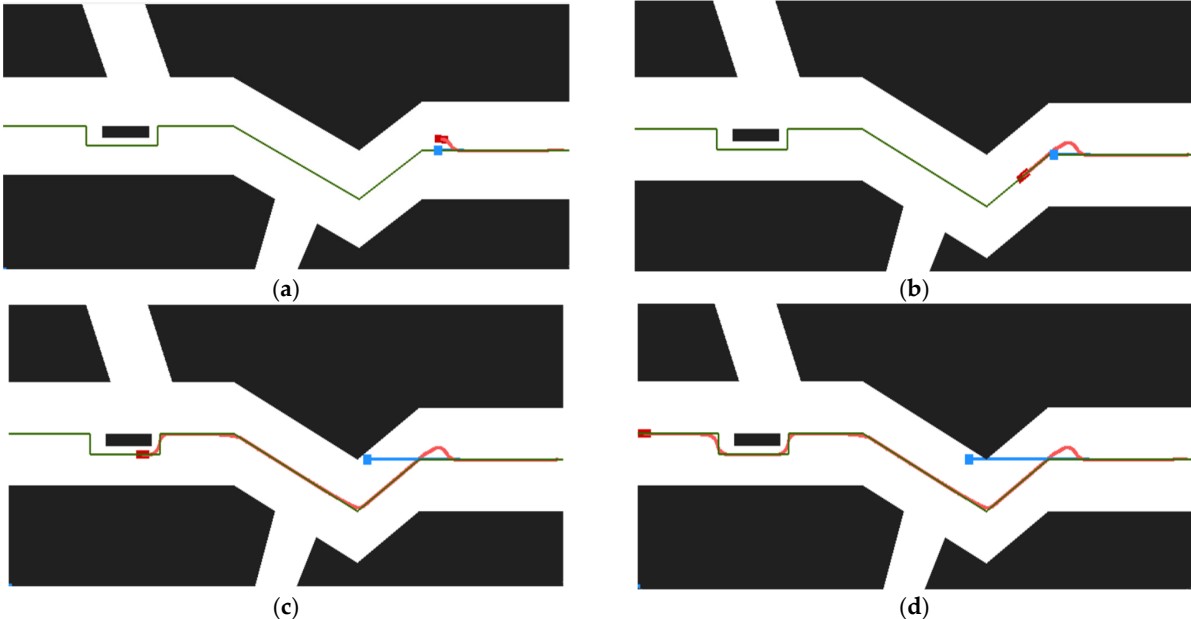

(**a**)　　　　　　　　　　　　　　　　(**b**)

(**c**)　　　　　　　　　　　　　　　　(**d**)

**Figure 16.** Tunnel detection local planning results. Note: ■: known environment; ■: unknown obstacles; ■: AUV; ——: the AUV's navigation path; ——: global planning path; ——: obstacle path.

The four stages of the AUV's local motion planning procedure are shown in Figure 16a–d. Figure 16a,b depict an AUV avoiding unknown obstacles, whereas Figure 16c,d depict an AUV traveling along the global path and arriving at the destination to finish the mission. The mode of motion of the obstacle is set to horizontal uniform linear motion with velocity 0.1 m/s horizontally to the left.

Based on the planning technique described in this work, it becomes evident that the AUV can effectively track the global path during its initial stages. When the AUV encounters an unknown obstacle, it promptly adjusts its heading based on the trained policy to avoid the obstacle, ensuring the safe completion of the AUV tunnel detection task. The errors and position changes that occur during the AUV's motion are illustrated in Figures 17 and 18 below.

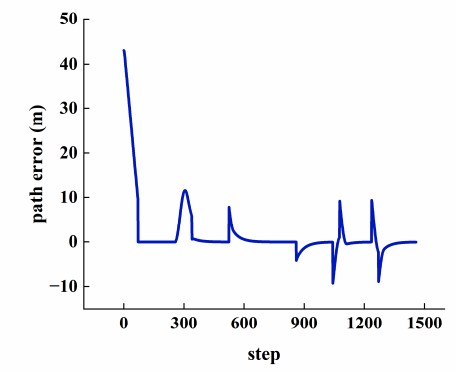

**Figure 17.** Global path following errors.

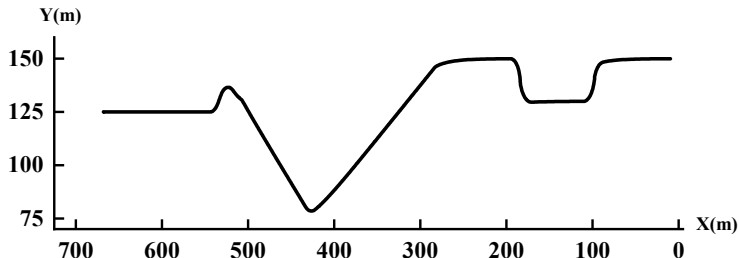

**Figure 18.** Position change during the tunnel motion of the AUV.

Combining the following error and position transformation, it can be seen that the part with large fluctuations in Figure 17 is the tracking error generated by the initial stage of the AUV and the sudden obstacle avoidance planning, and the part with small oscillations is the error generated by the constraint of the AUV's return angle when turning.

In summary, using the designed strategy, simulation experiments for AUV local motion planning have been conducted, which include the sensing–planning experiment, the planning–execution experiment, and the end-to-end local planning simulation experiment. The experiments successfully demonstrated the effectiveness, stability, and advantages of the proposed algorithm. Moreover, within the context of underwater tunnel detection, the entire process of the local motion planning architecture was implemented.

## 5. Conclusions

To tackle the challenges of sparse rewards, limited strategies, and inadequate environmental adaptability in a AUV's local motion planning task, this paper proposes an end-to-end method of perception–action–execution based on the double-layer deep deterministic policy gradient (DDPG) algorithm. The proposed method overcomes the high training requirements and learning difficulties associated with the direct thrust output end-to-end approach. During the validation phase, simulation experiments were conducted using the Python language. The simulation results demonstrate that the HER-DDPG-based planning approach achieves higher reward values and faster convergence. Additionally, the DDPG-based execution method exhibits stable and effective tracking results for various path tracking tasks while meeting the error constraints. Moreover, the designed end-to-end algorithm enables local motion planning in tunneling scenarios, a typical application, with the maximum tracking error being controlled within a manageable limit. Overall, this paper showcases the feasibility of the proposed theoretical framework.

In this paper, we introduced a creative integration of the traditional hierarchical planning process with end-to-end planning concepts to construct a comprehensive local motion planning architecture that encompasses sensing, planning, and control. By enabling information sharing and interaction among AUVs, this integration allows for a more cohesive integration of the vehicles, opening up new possibilities for local motion planning. This article presents several significant innovations and contributions, which can be summarized as follows:

- In response to the inherent challenge of reward sparsity in the local motion planning task of AUVs, a goal-directed hindsight experience replay method was introduced. This method effectively improves the stability of training, enhances sample efficiency, and reduces the interaction cost between AUVs and the environment.
- In order to tackle the challenge of the AUV's inadequate obstacle avoidance capabilities when confronted with dynamic obstacles, an obstacle avoidance function was devised utilizing the repulsive force protection domain within the virtual potential field. This function establishes the foundation for a dynamic obstacle avoidance mechanism specifically designed for the AUV. Furthermore, the reward function for the AUV's obstacle avoidance module was meticulously crafted, drawing inspiration from the obstacle avoidance function.

- A method for AUV execution was designed, leveraging the DDPG algorithm. This method utilizes the dynamics model of the AUV as the training environment to establish the mapping relationship between force and action. This control algorithm does not require the development of an accurate AUV model and improved control stability and robustness for the AUV.
- A localized end-to-end motion planning architecture was designed that integrates state information–action output–control force output. The primary objective of this architecture is to simplify the hierarchical motion planning process for AUVs. Simultaneously, it aims to reduce the challenges associated with traditional end-to-end planning and enhance the smoothness and stability of AUV motion.

Although this paper makes good progress in AUV motion planning methods by incorporating artificial intelligence methods, the research in this paper still has some limitations, as well as issues that still need to be investigated in depth in the future:

- High-dimensional state space: The state space in this study only includes the position and the sonar detection information, and the dimension of the state space is not very high. If the state space of the robot is very high-dimensional and multiple sensor information or motion parameters need to be considered, then the planning complexity increases significantly, resulting in lower computational efficiency.
- Lack of practical validation: Due to cost and time, testing personnel, and other limitations, out-of-field experiments were not conducted for validation in this study. Performing off-site experiments in real-world scenarios is crucial for data collection, model validation, and algorithm validation and improvement, which helps ensure that AUVs are able to perceive, plan, and perform tasks correctly. The method presented in this paper needs to be further improved based on future off-site experiments if it is to be practically applied to the inspection task of AUV.
- Three-dimensional motion planning: Considering the common scenarios of AUV operation, to simplify this study, we decoupled the AUV motion and only considered the motion in the horizontal plane. However, an AUV is a six-DoF robot, and fully considering its six DoF for motion planning is also a direction for further research. Therefore, future research on AUV 3D motion planning is needed.

**Author Contributions:** Conceptualization, Y.S.; methodology, X.L.; validation, X.L.; investigation, L.W.; resources, Y.S.; data curation, J.T.; writing—original draft preparation, X.L.; writing—review and editing, X.L.; visualization, X.L. and J.T. and L.Z.; supervision, Y.S.; project administration, L.W.; funding acquisition, Y.S. All authors have read and agreed to the published version of the manuscript.

**Funding:** This research was funded by the Natural Science Foundation of Heilongjiang Province of China (grant number ZD2020E005), and the National Natural Science Foundation of China (grant number 52071104).

**Institutional Review Board Statement:** Not applicable.

**Informed Consent Statement:** Not applicable.

**Data Availability Statement:** Not applicable.

**Acknowledgments:** The author would like to thank the reviewers for their comments on improving the quality of this paper.

**Conflicts of Interest:** The authors declare no conflict of interest.

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
