# Peer review of "End-to-End AUV Local Motion Planning Method Based on Deep Reinforcement Learning"

_jmse, doi:10.3390/jmse11091796_

Round 1

Reviewer 1 Report

The authors have proposed a deep RL based motion planning solution for AUV. The work is backed by substantial results which are interesting. However, there are several issues that need to be addressed.

1. The abstract says the motion planning is connected with control; however, the subsequent sections do not make it clear.

2. The work talks about constraint control. Nevertheless, when a priori bounds on uncertainty is known, it is possible to design control for marine vessels and literature review needs to be improved on this aspect (see "A switching control perspective on the offshore construction scenario of heavy-lift vessels", "Robustifying dynamic positioning of crane vessels for heavy lifting operation" and similar others).

3. It is not understood why modelling is included, but it is not used in the proposed design, at least explicitly.

4. The experiment section is nothing but simulation, so why it is shown as a separate section.

5. Comparison with existing methods is necessary. 

Nothing major to comment.

Reviewer 2 Report

- Applying deep RL to AUV in simulation is not new. Therefore, I would urge the authors to have the experimental results (be these of preliminary nature). This is important to advance the research in the subject domain on AUVs.

- Another way to add further novelty in the work is to improve the obstacle avoidance algorithm so that it is capable of dealing a dynamic environment.

- Please explicitly mention the limitations of the proposed motion planning method?

- What are the assumptions involved in the study? Please mention these and also include the real-world implications of these assumptions?

- Figure 9 contains relatively simple and straight-forward goal-directed behavior simulation experiments. Please include more complex scenarios commonly encountered in a real-world situation.

- Redraw Figure 8 using distinguishing markers. Plot one quantity with solid line while other quantity with dotted line.

Moderate English language changes required.

Reviewer 3 Report

1.    In page 6, Section 2.2, please specify the number of the figure instead of using following figure. It can be confused just mentioning ‘the following figure’.

2.    Also the figure 2 requires coordinate system to figure out the pose of it.

3.    The authors mention ‘A’ as the discounted reward total return in the second line of page 9, but the related equation is not shown in the text, so it needs to be described. In addition, it is described that ‘A’ mentioned after equation (10) means an update coefficient. Please clarify both two ‘A’s.

4.    What are A, B, C, D, and E? Where these are presented? It is difficult to understand because the leap in explanation is so severe. Explain with detailed explanations and figures or equations Also, at the bottom of page 11, the authors used another ‘A’ as the set of action. Throughout the paper, a single character should be used for one meaning or purpose. Please, correct it.

5.    In this paper, even though it is an autonomous underwater vehicle, it creates and controls a path on a 2D plane with 2D sensing information. How is it different from the existing unmanned ship control? In other words, a clear description of the differentiation and contribution of this study is required.

It is good.

Reviewer 4 Report

In general, the paper is well-written, organized, and of an appropriate length. While my overall opinion of this paper is positive, there are certain areas where the content could be clarified. To improve the quality of the article, I recommend considering the following suggestions:

What research gap did you find from previous researchers in your field (it is still partially described, but needs to be expanded and made clearer)? The main contributions of the manuscript are not clear. I suggest clarify and expand the description of main contributions. It will improve the strength of the article

In the results presented, the authors do not include external disturbances in the dynamic model given in equation (3). Furthermore, they assume that the term g₀ is zero. To evaluate the developed algorithms in a more realistic scenario, the authors could include numerical tests where external disturbances are considered.

In the results presented, the neural network was built using Pytorch and the model was trained using a GPU. In Section 4, Simulations, it seems that there is not enough description of how the neural network was built and the description of the hardware. It is expected to show more details of the simulation process.

Round 2

Reviewer 1 Report

No further comments.

No further comments.

Reviewer 2 Report

I understand that it might not be possible to include the experimental results due to the need of having a complicated hardware setup, however, at least, my comment suggested earlier (improve the obstacle avoidance algorithm so that it is capable of dealing a dynamic environment) needs to be addressed. You can not put everything in future work...

Moderate changes required.

Reviewer 3 Report

All the comments are answered well. And I think this version of the paper can be published.

Reviewer 4 Report

The authors successfully answer to my concerns. In my opinion, the manuscript can be accepted in its present form.

Round 3

Reviewer 2 Report

Authors have addressed the suggested comments. 

Moderate editing of English language required